# DIFFERENTIALLY PRIVATE BIAS-TERM FINE-TUNING OF FOUNDATION MODELS

## ABSTRACT

We study the problem of differentially private (DP) fine-tuning of large pre-trained models – a recent privacy-preserving approach suitable for solving downstream tasks with sensitive data. Existing work has demonstrated that high accuracy is possible under strong privacy constraint, yet requires significant computational overhead or modifications to the network architecture.

We propose differentially private bias-term fine-tuning (DP-BiTFiT), which matches the state-of-the-art accuracy for DP algorithms and the efficiency of the standard BiTFiT. DP-BiTFiT is model agnostic (not modifying the network architecture), parameter efficient (only training about $0.1\%$ of the parameters), and computation efficient (almost removing the overhead caused by DP, in both the time and space complexity). On a wide range of tasks, DP-BiTFiT is $2 \sim 30\times$ faster and uses $2 \sim 8\times$ less memory than DP full fine-tuning, even faster than the standard full fine-tuning. This amazing efficiency enables us to conduct DP fine-tuning on language and vision tasks with long-sequence texts and high-resolution images, which were computationally difficult using existing methods.

## 1 INTRODUCTION

Fine-tuning from large pre-trained neural networks is one of the most critical technique in deep learning, yielding strong performance in a variety of domains (Pan & Yang, 2009; Kenton & Toutanova, 2019; Goyal et al., 2017). Among different methods, full fine-tuning is the most prevalent one, which trains all the model parameters on the downstream tasks and achieves high accuracy within a small number of training epochs. However, full fine-tuning on large models, from hundreds of millions (He et al., 2016; Chen et al., 2016) to billions of parameters (Brown et al., 2020), can be burdensome in terms of the computation and the deployment, since a full copy of fine-tuned model parameters is needed for each task.

To alleviate this issue, the parameter efficient fine-tuning only trains a substantially small portion of the model parameters, in contrast to the full fine-tuning. At a high level, the parameter efficient fine-tuning methods can be divided into two categories. $\langle 1 \rangle$ Model-aware methods, meaning a relatively small number of parameters are introduced into the neural network architecture and only the new parameters are optimized. Examples include LoRA (Hu et al., 2021), Adapter (Houlsby et al., 2019), and Compacter (Mahabadi et al., 2021). $\langle 2 \rangle$ Model-agnostic methods, meaning that only a subset of existing parameters are trainable. Examples include training only the output linear layer (also known as the classification head), (Kornblith et al., 2019)), only the layer normalization layer (Houlsby et al., 2019) and bias-term fine-tuning (BiTFiT) (Zaken et al., 2022). We illustrate the differences in Equation (1): $\mathbf{W}_0, \mathbf{b}_0$ are the pre-trained weights and biases, ' $\hat{}$ ' indicates trainable parameters, and $\boldsymbol{\theta}$ is the additional parameters.

$$\underbrace{f(\boldsymbol{x}; \mathbf{W}_0, \mathbf{b}_0)}_{\text{pre-trained model}} \longrightarrow \underbrace{f(\boldsymbol{x}; \hat{\mathbf{W}}, \hat{\mathbf{b}})}_{\text{full fine-tuning}} \quad \text{or} \quad \underbrace{f(\boldsymbol{x}; \mathbf{W}_0, \mathbf{b}_0, \hat{\boldsymbol{\theta}})}_{\text{model-aware fine-tuning}} \quad \text{or} \quad \underbrace{f(\boldsymbol{x}; \mathbf{W}_0, \hat{\mathbf{b}})}_{\text{bias-term fine-tuning}} \quad (1)$$

Empirically, these parameter efficient fine-tuning methods have achieved high accuracy that is comparable to the full fine-tuning in the standard non-private setting. For instance, last-layer training (also known as linear probing) of ResNet (He et al., 2016) and Vision Transformer (ViT, (Dosovitskiy et al., 2020)) achieves 80% accuracy on the ImageNet dataset (Sun et al., 2017; Kornblith et al., 2019); LoRA and BiTFiT of RoBERTa (Liu et al., 2019) and BERT (Kenton & Toutanova, 2019)

achieve about 94% on SST2, 87% on MNLI, and on average 85% across the General Language Understanding Evaluation (GLUE) datasets (He et al., 2021; Hu et al., 2021). In addition, parameter efficient methods are faster than full fine-tuning and save the communication cost significantly in the distributed learning.

Parallel to these developments, the success of deep learning models relies on the availability of large datasets, which may contain sensitive information to be protected rigorously. This privacy issue is well-known for neural networks can be vulnerable to privacy attacks: membership information can be leaked from the purchase records via Google and Amazon online services (Shokri et al., 2017); sensitive texts can be reconstructed by specifically designed prefix on GPT2 (Carlini et al., 2021) and so can images in CIFAR10 and MNIST (Haim et al., 2022). To protect against such privacy risks, the standard technique is differential privacy (DP, formally stated in Definition 2.1), which randomizes the standard optimizers by updating with the private gradient in Equation (2).

A recent line of work has extensively studied the DP fine-tuning in both computer vision and language tasks, often achieving less than 3% accuracy drop across different settings via full fine-tuning (De et al., 2022; Li et al., 2021; Bu et al., 2022b;a), last-layer (Mehta et al., 2022), LoRA, Adapter, or Compacter (Yu et al., 2021a). In fact, fine-tuning or pre-training from large dataset is considered necessary in the DP deep learning literature. As a matter of fact, full fine-tuning DP-GPT2 only achieves 24.2 BLEU score ($\epsilon = 8$) on E2E dataset if randomly initialized (Li et al., 2021), in starking contrast to 63.2 BLEU if pre-trained; similarly, state-of-the-art (SOTA) DP accuracy on ImageNet is 48% ($\epsilon = 10$) without pre-training (Kurakin et al., 2022) but 86.7% accuracy if pre-trained (De et al., 2022). Specifically, parameter efficient DP fine-tuning has empirically demonstrated strong accuracy (see our Table 3) with $3 \sim 4\times$ memory saving and $2 \sim 3\times$ speedup compared to DP full fine-tuning by Opacus (c.f. Figure 3 and Yu et al., 2021a, Table 3). Although previous works have shed light on various DP fine-tuning methods, we are the first to study DP-BiTFiT specifically and to show two distinctive advantages of it.

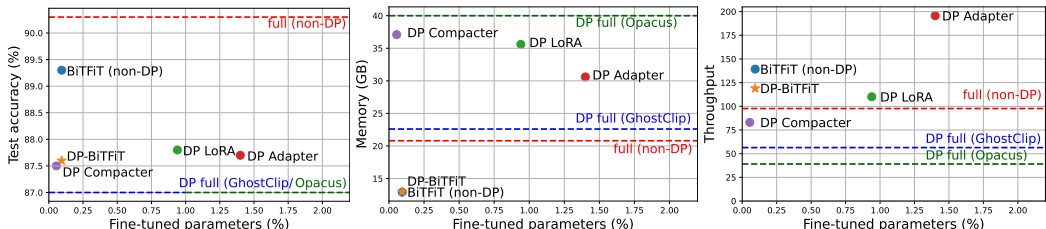

Figure 1: Performance of different fine-tuning methods on MNLI dataset with RoBERTa-large. Note that DP-BiTFiT has a dominating advantage on memory-saving (requiring less than half the memory of other methods), and is on par with the fastest and the most accurate DP fine-tuning method.

Firstly, DP-BiTFiT is model-agnostic and remains its parameter efficiency around 0.1% across models by Table 1. While last-layer training is also model-agnostic, it performs poorly on transformers (Li et al., 2021) and could be parameter inefficient on multi-class tasks (e.g. last-layer needs to train 8% parameters of ResNet50 on ImageNet with 1000 classes). Similarly, LoRA, Adapter and Compacter are architecture-dependent (e.g. mostly restricted to transformers but not applicable to ResNet), and possibly parameter inefficient (e.g. LoRA and Adapter may need to train more than 12% parameters of BART-large (Lewis et al., 2020) to achieve high accuracy by (He et al., 2021, Figure 1& 4)). These characteristics make them difficult to directly apply on general network architectures.

Secondly, DP-BiTFiT is computationally efficient, almost as much as the standard BiTFiT and significantly more efficient than DP full fine-tuning, particularly with large models and high-dimensional input data. For examples of DP full fine-tuning, (Li et al., 2021) have reported $2 \sim 4\times$ slowdown on large language models for four advanced private codebases and up to $5\times$ memory overhead, compared to the standard fine-tuning; even on small networks, 11 codebases across Tensorflow, JAX, and Pytorch have demonstrated $0.2 \sim 5\times$ slowdown and $3 \sim 100\times$ reduction in maximum batch size in (Subramani et al., 2021). See more discussion in Section 3.3.

**Contributions.** In this work, we develop DP-BiTFiT, a fine-tuning method that is model-agnostic, accurate, privacy-preserving, parameter efficient, and computationally efficient.

1. Algorithmically, we propose the Differentially Private Bias-Term Fine-Tuning (DP-BiTFiT) in Algorithm 1 that is highly accurate under DP constraint, on par with SOTA in Section 4 and even outperforming fully fine-tuned GPT2-large.

2. DP-BiTFiT is model-agnostic and only optimizes 0.1% of the model parameters on BERT, RoBERTa, GPT2, ViT, ResNet, and so on (see Table 1). Thus DP-BiTFiT is one of the most *parameter efficient* fine-tuning methods among DP LoRA, Adapter, last-layer, etc.

3. We design a *computationally efficient* implementation of DP-BiTFiT, whose time and space complexity is almost the same as the standard non-DP BiTFiT, while being faster than non-DP full fine-tuning and other DP fine-tuning (see Figure 1). This advantage is analyzed in Table 2, and demonstrated via the substantial speedup and memory-saving in Figure 3 and Figure 4.

4. DP-BiTFiT is a unique algorithm in that *the computation overhead is independent of the feature dimension $T$[1]*. This is due to the *activation-free forward pass* that only happens in the no-weight training[2] unlike LoRA. In Figure 1, although DP-BiTFiT optimizes a similar number of parameters to DP LoRA or Compacter, its memory efficiency is dominating. Therefore, DP-BiTFiT enjoys a special advantage when applied on long-sequence texts and high-resolution images (see Figure 3).

**Novelty.** At a glance, our results may appear to be incremental as we are merely adding differential privacy to an existing method (BiTFiT) through a standard mechanism (DP-SGD). This is not true! Computationally, our implementation of DP-BiTFiT involves substantial algorithmic innovation (orthogonal to GhostClip (Goodfellow, 2015; Li et al., 2021) which *only works on the weights*, not the biases) that exploits the special structures in the forward and backward passes, hence removing the computational and memory overhead in DP-SGD, which can be unavoidable in other methods.

---

**Algorithm 1** Bias-Term Fine-Tuning (BiTFiT) v.s. DP-BiTFiT

---

**Parameters:** $l$-th layer's bias $\mathbf{b}_l$, subsampling probability $p$, number of iterations $T$, number of layers $L$, noise scale $\sigma$, clipping threshold $R$.

1: **for** iteration $t = 1, \cdots, T$ **do**
2:    Subsample a batch $B_t \subseteq \{1, \ldots, n\}$ from training set with probability $p$
3:    **for** layer $l \in L, L-1, \cdots, 1$ **do**
4:        Get output gradient $\frac{\partial \mathcal{L}}{\partial \boldsymbol{s}_l}$
5:        Compute per-example gradient and its norm: $\frac{\partial \mathcal{L}_i}{\partial \mathbf{b}_l} = \frac{\partial \mathcal{L}}{\partial \boldsymbol{s}_{l,i}}^\top \mathbf{1} \implies \|\frac{\partial \mathcal{L}_i}{\partial \mathbf{b}_l}\|_F^2$
6:        Aggregate gradient norms across all layers: $\|\frac{\partial \mathcal{L}_i}{\partial \mathbf{b}}\|_F^2 = \sum_l \|\frac{\partial \mathcal{L}_i}{\partial \mathbf{b}_l}\|_F^2$
7:        Compute clipping factor: $C_i = C(\|\frac{\partial \mathcal{L}_i}{\partial \mathbf{b}}\|_F; R)$
8:        Compute sum of clipped gradients $\mathbf{G} = \sum_i C_i \frac{\partial \mathcal{L}_i}{\partial \mathbf{b}}$ (note $C_i = 1$ if in standard BiTFiT)
9:        Add Gaussian noise $\mathbf{G} = \mathbf{G} + \sigma R \cdot \mathcal{N}(0, \mathbf{I})$
10:       Descend on bias terms with the gradient $\mathbf{G}$ by SGD/Adam/...

---

## 2 PRELIMINARIES

**Fine-tuning methods.** Fine-tuning, i.e. training a model on a large dataset for a sufficiently long time, and then continuing to train (or transferring) onto the downstream datasets, is the standard paradigm to achieve high accuracy in both the standard and the DP regimes. In DP deep learning, the pre-training takes place on a public dataset using regular optimizers like SGD, and the fine-tuning takes place on a private dataset which requires privacy protection, using DP optimizers like DP-SGD in Section 2.

In a long line of research, various fine-tuning methods have been proposed. One of the most popular method is the full fine-tuning, which simply runs gradient descents on all trainable weights and

---

[1]As summarized in Table 2 and Table 7, the computation overhead to get the per-sample weight gradient norm is linear (by instantiating per-sample gradints) or quadratic in $T$ (if using the ghost norm trick (Goodfellow, 2015; Li et al., 2021)), for DP full and parameter efficient fine-tuning.

[2]We distinguish the weight training and bias training in Section 2 using the chain rules. Note that activation-free means memory-saving, which is not leveraged by DP full, LoRA, Adapter, Compacter, etc.

biases, thus can be inefficient when the model is large. To improve the efficiency, (Li & Liang, 2021) proposes the prefix tuning that only optimizes the prompts or the input layer activation (Lester et al., 2021; Liu et al., 2021). However, as pointed out in (Hu et al., 2021) and (Li et al., 2021), the prefix tuning can be difficult to optimize and thus sub-optimal on large models. Another approach is to reduce the number of trainable parameters. For example, LoRA (Hu et al., 2021), Adapter (Houlsby et al., 2019; Rebuffi et al., 2017; Pfeiffer et al., 2021; Rücklé et al., 2021; Lin et al., 2020) and Compacter (Mahabadi et al., 2021) insert small 'adapter' layers (usually 1-10% of total parameters) between existing layers, and only the newly added adapters are optimized. We describe the forms of LoRA and Adapter in Appendix C and analyze their complexity.

In addition to the aforementioned methods, BiTFiT is a special parameter-efficient method that rivals the full fine-tuning (Zaken et al., 2022; Cai et al., 2020; He et al., 2021). Firstly, BiTFiT optimizes a subset of original parameters – the bias terms, which usually constitute less than 1/1000 of all parameters as demonstrated in Table 1. Therefore, BiTFiT can be readily deployed to any network in a model-agnostic manner. Secondly, BiTFiT is fundamentally different to other parameter efficient methods such as LoRA, since the bias gradients are computed differently than the weight gradients on the computation graph. We will elaborate on this in Equation (4).

**Deep learning with differential privacy.** We recall the classic $(\epsilon, \delta)$-DP, under which we train deep neural networks with provably privacy guarantees.

**Definition 2.1** ((Dwork et al., 2006)). A randomized algorithm $M$ is $(\varepsilon, \delta)$-differentially private if, for any two neighboring datasets $S, S'$ that differ by one datapoint and for any event $E$, we have $\mathbb{P}[M(S) \in E] \leqslant e^{\varepsilon} \mathbb{P}[M(S') \in E] + \delta$.

In deep learning, DP can be achieved through applying an off-the-shelf optimizer (SGD or Adam) with a privately released stochastic gradient in place of the regular $\sum_i \boldsymbol{g}_i$. The private stochastic gradient is computed by first getting a minibatch $\mathcal{I}$ via Poisson sampling, then compute

$$\text{Private gradient} \quad \sum_{i \in \mathcal{I}} \boldsymbol{g}_i \cdot C(\|\boldsymbol{g}_i\|; R) + \sigma R \cdot \mathcal{N}(0, \mathbf{I}), \tag{2}$$

where $C$ is any function[3] $\mathbb{R}^+ \to \mathbb{R}$ subject to $C(x) \leq R/x$, $\boldsymbol{g}_i$ is the $i$-th per-sample gradient, $R$ is the clipping threshold, and $\sigma$ is the noise multiplier. The private gradient is guaranteed to be DP through the *sampled-Gaussian mechanism* and the associated tight privacy accounting to compose over the iterations (see, e.g., Abadi et al., 2016; Wang et al., 2019; Mironov et al., 2019; Koskela et al., 2020; Bu et al., 2020; Gopi et al., 2021, and the references therein.).

**Backward propagation.** We briefly introduce the back-propagation, which reveals a simple yet important difference between the gradients of weights and those of biases. We consider a linear layer, indexed as the $l$-th layer, with weight $\mathbf{W}_l \in \mathbb{R}^{d \times p}$ and bias as $\mathbf{b}_l \in \mathbb{R}^p$. We leave the derivation of other layers such as normalization and convolution in Appendix A.1. We denote the mini-batched input of this layer as $\boldsymbol{a}_l \in \mathbb{R}^{B \times T \times d}$ and the immediate output as $\boldsymbol{s}_l \in \mathbb{R}^{B \times T \times p}$, where $B$ is the batch size and $T$ is the feature dimension[4]: $\boldsymbol{a}_{l+1} = \phi(\boldsymbol{s}_l), \boldsymbol{s}_l = \boldsymbol{a}_l \mathbf{W}_l + \mathbf{b}_l$. Here $\phi$ is any non-parametric inter-layer operation, e.g. the non-linear activation (like ReLU), pooling, padding, and so on.

We write $\mathcal{L} = \sum_{i=1}^{n} \mathcal{L}_i$ as the total loss ($n$ being total sample size) and $\mathcal{L}_i$ as the per-sample loss of the $i$-th sample. During a standard back-propagation of $L$ layers, the chain rule keeps track of the *output gradient* at each layer in a just-in-time fashion:

$$\frac{\partial \mathcal{L}}{\partial \boldsymbol{s}_l} = \frac{\partial \mathcal{L}}{\partial \boldsymbol{a}_L} \circ \frac{\partial \boldsymbol{a}_L}{\partial \boldsymbol{s}_{L-1}} \cdot \frac{\partial \boldsymbol{s}_{L-1}}{\partial \boldsymbol{a}_{L-1}} \circ \cdots \frac{\partial \boldsymbol{a}_{l+1}}{\partial \boldsymbol{s}_l} = \frac{\partial \mathcal{L}}{\partial \boldsymbol{s}_{l+1}} \mathbf{W}_{l+1} \circ \phi'(\boldsymbol{s}_l). \tag{3}$$

This output gradient $\frac{\partial \mathcal{L}}{\partial \boldsymbol{s}_l}$ is used to compute per-sample gradient of weights and biases,

$$\frac{\partial \mathcal{L}_i}{\partial \mathbf{W}_l}^{\top} = \sum_j \frac{\partial \mathcal{L}_i}{\partial \boldsymbol{s}_{l,j}}^{\top} \frac{\partial \boldsymbol{s}_{l,j}}{\partial \mathbf{W}_l} = \frac{\partial \mathcal{L}}{\partial \boldsymbol{s}_{l,i}}^{\top} \boldsymbol{a}_{l,i}, \quad \frac{\partial \mathcal{L}_i}{\partial \mathbf{b}_l}^{\top} = \sum_j \frac{\partial \mathcal{L}_i}{\partial \boldsymbol{s}_{l,j}}^{\top} \frac{\partial \boldsymbol{s}_{l,j}}{\partial \mathbf{b}_l} = \frac{\partial \mathcal{L}}{\partial \boldsymbol{s}_{l,i}}^{\top} \mathbf{1}. \tag{4}$$

Notably, the weight gradient needs the activation tensor $\boldsymbol{a}_l$ to compute an expensive $O(BTpd)$ tensor multiplication. Memory-wise, $\{\boldsymbol{a}_l\}_l$ across all layers is very costly to store (see Footnote 5). In

---

[3]Examples of gradient clipping include but not limited to Abadi's clipping $\min(R/\|\boldsymbol{g}_i\|, 1)$ (Abadi et al., 2016) and automatic clipping (AUTO-S) $R/(\|\boldsymbol{g}_i\| + 0.01)$ (Bu et al., 2022b; Yang et al., 2022).

[4]In sequential data such as text, $T$ is the sequence length; in vision data, $T$ is the product of input dimensions (e.g. for images, $T$ is the product of height and width). We refer to a high-dimensional input when $T$ is large.

sharp contrast, the computation of bias gradient does not need $\boldsymbol{a}_l$, and the multiplication with $\mathbf{1}$ in Equation (4) is actually a cheap $O(BTp)$ summation on $\frac{\partial \mathcal{L}}{\partial \boldsymbol{s}_l} : B \times T \times p \to B \times p$.

**Forward propagation.** During the forward propagation, all codebases for DP algorithms such as Opacus, Private Transformers and others (Yu et al., 2021a; Bu et al., 2022a) need to compute the activation tensors $\{\boldsymbol{a}_l\}_l$ for all layers inside the computation graph, to be used in equation 4 at high memory cost[5]. Especially for huge models like GPT3 (Brown et al., 2020) with 175B parameters, the memory burden incurred by the activation grows extremely large: the activation tensors $\boldsymbol{a}_l$ consume more than 3600GB of memory while the parameters and the gradients only consume 300GB (Rajbhandari et al., 2020). On one hand, this issue can be alleviated by the activation recomputation or checkpointing technique (Chen et al., 2016; Jain et al., 2020), whose memory cost reduces from $O(L)$ to $O(\sqrt{L})$ with an unfortunate 33% slowdown. Alternatively, we note that the activation tensors are not necessary in the forward propagation, if we only optimize the bias terms.

## 3   DIFFERENTIALLY PRIVATE BIAS-TERM FINE-TUNING

We propose DP-BiTFiT, to privately train only the bias terms in a neural network by combining Equation (4) and Equation (2). We use shaded lines to represent the additional DP operations in Algorithm 1, and add DP-related variables and operations in red in the computation graph by Figure 2.

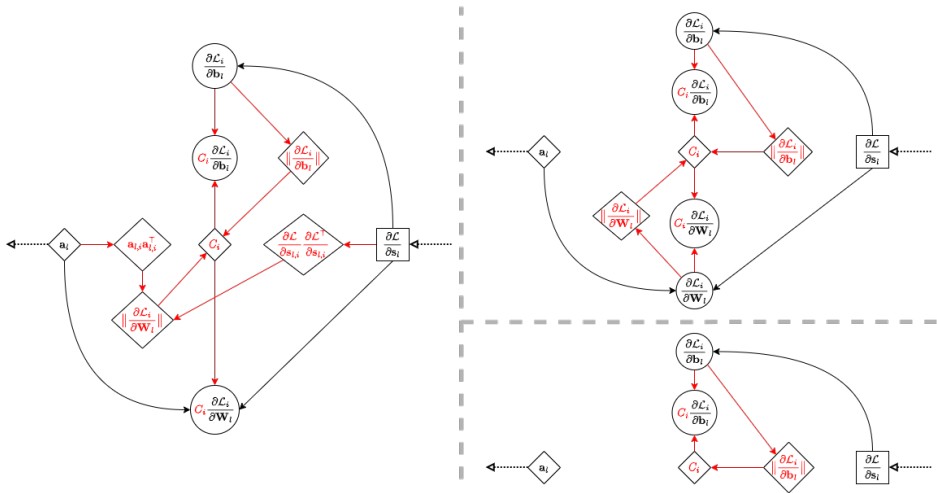

Figure 2: Back-propagation for DP (red&black) and non-DP (black) algorithms. Left: full fine-tuning with GhostClip (ghost clipping; (Goodfellow, 2015; Li et al., 2021; Bu et al., 2022a)). Upper right: full fine-tuning with Opacus (Yousefpour et al., 2021). Lower right: BiTFiT.

Implementation-wise, DP-BiTFiT is different from all existing DP algorithms (including full, LoRA, Adapter, etc.) that optimize weights, since it does not apply a Pytorch forward hook to store the activation $\boldsymbol{a}_l$ for all layers. We provide the implementation details of DP-BiTFiT in Appendix B. To give a concrete example, we apply DP-BiTFiT to the RoBERTa-large model on QQP dataset, following the same setting as (Li et al., 2021) and using one 40GB A100 GPU. This is the most time-consuming text classification task in our work, taking 119 minutes per epoch for a training batch size 20 using the fastest DP full fine-tuning implementation – GhostClip (Li et al., 2021). To conduct a simple ablation study, setting all weights to not require gradients (but forward hooks are still operating) reduces the training time by 50% to to 80 minutes; removing the forward hooks further reduces the training time by 30% to 63 minutes; finally, using the maximum batch size allowed by the memory-saving DP-BiTFiT reduces to 43 minutes.

---

[5]Across different models including VGG, ResNet, DenseNet, and RoBERTa, the activation tensors can take more than 95% memory of the entire training (see (Jain et al., 2020, Figure 3)).

### 3.1 PARAMETER EFFICIENCY

DP-BiTFiT enjoys exactly the same parameter efficiency as the standard BiTFiT, training merely about 0.1% of the total parameters in large models. We demonstrate that DP-BiTFiT is one of the most parameter-efficient fine-tuning through a list of models in Table 1, extended in Table 11.

An advantage of this parameter efficiency is reflected in the computation efficiency, given that most parameters do not require gradients to be computed: we show in Table 2 and Section 3.3 that DP-BiTFiT is much more efficient than full fine-tuning (DP and even non-DP). Additionally, the parameter efficiency also translates to the communication efficiency in the distributed learning. For example, the 64-bit communication cost of DP full fine-tuning is $64MD$ where $M$ is number of worker and $D$ is total number of parameters, which can be improved to $0.064MD$ by DP-BiTFiT.

Table 1: Parameter efficiency of (DP) BiTFiT.

| Dataset | Model | # of params | % of params |
|---|---|---|---|
| ImageNet | VGG16 | 138M | 0.009 |
| | ResNet18 | 11.7M | 0.043 |
| | ResNet50 | 25.6M | 0.113 |
| | ViT-small-patch16 | 21.7M | 0.238 |
| | ViT-base-patch16 | 85.8M | 0.120 |
| | ViT-large-patch16 | 303M | 0.090 |
| E2E | GPT2-small | 124M | 0.082 |
| | GPT2-medium | 355M | 0.076 |
| | GPT2-large | 774M | 0.066 |
| GLUE | RoBERTa-base | 125M | 0.083 |
| | RoBERTa-large | 355M | 0.077 |

### 3.2 COMPLEXITY OF WEIGHT AND BIAS TRAINING

We present in Table 2 the complexity of DP training on weights and biases, for one layer mapping $B \times T_l \times d_l$ to $B \times T_l \times p_l$. To elaborate on Footnote 4, for text data, $T_l$ is the sequence length, $d_l$ is input dimension, and $p_l$ is output dimension; for image data and specially in a convolution layer, $T_l$ is height times width, $d_l$ is the input channels times kernel sizes, $p_l$ is the output channels (c.f. Bu et al., 2022a, Section 2.3). Notice that the total complexity of training a network is summed across all layers, e.g. the time complexity of standard full training is $6B \sum_l T_l p_l d_l$, DP full fine-tuning is over $8B \sum_l T_l p_l d_l$, and DP-BiTFiT is about $4B \sum_l T_l p_l d_l$. Therefore, our complexity analysis indicates that DP-BiTFiT is $6/4 = 1.5\times$ faster than non-private full fine-tuning and over $8/4 = 2\times$ faster than DP full fine-tuning.

Table 2: Per-layer time and space complexity of training on weights (full fine-tuning) and biases. '$+$' means additional overhead to non-DP training, and '$\langle \rangle$' means between two values.

| | forward &output grad | weight training | | | | bias training | |
|---|---|---|---|---|---|---|---|
| | | non-DP | Opacus | GhostClip | MixGhostClip | non-DP | DP (ours) |
| Time complexity | $4BTpd$ | $2BTpd$ | $+2BTpd$ | $+2BTpd$ $+2BT^2(p+d)$ | $+2BTpd$ $+\langle 2BT^2(p+d), 2BTpd\rangle$ | $BTp$ | $+3Bp$ |
| Space complexity | $pd+$ $BT(p+d)$ | $BT(p+d)$ | $+Bpd$ | $+2BT^2$ | $+\min\{2BT^2, 2Bpd\}$ | $p$ | $+Bp$ |
| # back-prop | | 1 | 1 | 2 | 2 | 1 | 1 |
| forward hook | | ✗ | ✓ | ✓ | ✓ | ✗ | ✗ |

Here, the DP weight training (full fine-tuning) uses three efficient implementations that are equivalent mathematically but have different complexity: Opacus (Yousefpour et al., 2021), GhostClip (Goodfellow, 2015; Li et al., 2021), and MixGhostClip (Bu et al., 2022a). The first two implementations are illustrated in Figure 2, of which MixGhostClip is a hybridization that reduces to GhostClip when $T$ is small. These implementations have been thoroughly analyzed in (Bu et al., 2022a, Appendix C), and we take the complexity result from (Bu et al., 2022a, Table 1). For the complexity of bias training in Table 2, it suffices to analyze Line 5 of Algorithm 1. We refer the interested readers to Table 7 for details, where we also apply the complexity analysis of weight training on other methods beyond full fine-tuning, including DP LoRA and DP Adapter.

### 3.3 SCALABILITY OF DP ALGORITHMS

From the complexity analysis in Table 2, we observe that DP training on weights can be memory costly, especially when the models are large and the data is high-dimensional. As an example of the large modelling issue, (Li et al., 2021) shows that Opacus cannot fit even a single datapoint into a 16GB GPU using GPT2-large (Radford et al.) with 774M parameters, due to its $O(B \sum_l p_l d_l)$ space complexity where the number of parameters is $\sum_l p_l d_l$; for high-dimensional data, GhostClip cannot fit a single $400 \times 400$ image into the same GPU using ResNet18 with 11.7M parameters, due to its

$O(B \sum_l T_l^2)$ space complexity. Although MixGhostClip (Bu et al., 2022a) significantly alleviates the memory issue in both cases, it does so at a cost of roughly $2\times$ slowdown than the standard full fine-tuning (c.f. Bu et al., 2022a, Figure 4). In sharp contrast, DP-BiTFiT is amazingly scalable since its computational overhead is negligible and independent of $T$ (though the total complexity, mainly due to forward and output gradient, is still linear in $T$).

**Efficiency of DP training v.s. feature dimension** To empirically evaluate the computation efficiency of DP fine-tuning methods, we measure the time and GPU memory for a fixed batch size. We depict the high-dimensional data issue in Figure 3, in which the memory saving and speedup by DP-BiTFiT is substantial. We expect to observe greater efficiency advantage of DP-BiTFiT on higher dimensional data, e.g. in LLAMA2 (Touvron et al., 2023) and GPT4 (OpenAI, 2023) with $T = 4096$, in document-level language tasks with $T \approx 20000$ by (Beltagy et al., 2020), and in high-resolution image tasks, such as $1024 \times 1024$ CelebA-HQ (Karras et al., 2018) and Flickr-Faces-HQ (Karras et al., 2019) where $T$ can be of order $10^5$ in the convolution layers.

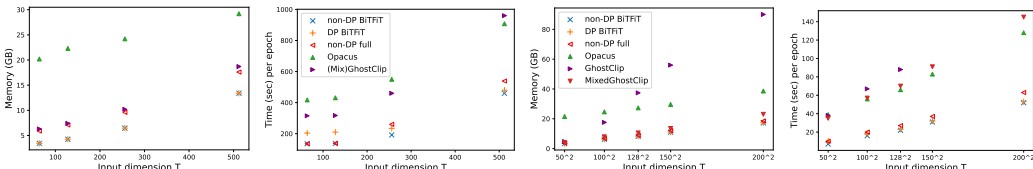

Figure 3: Memory and speed by different fine-tuning methods. Left two: SST2 dataset (sequence length $T$; MixGhostClip is equivalent to GhostClip for this small $T$) with RoBERTa-base and batch size 20. Right two: 50000 images of $\sqrt{T} \times \sqrt{T}$ pixels with ResNet50 and batch size 200.

**Efficiency of DP training v.s. model size** To stress-test the computation efficiency of DP-BiTFiT with large models, we apply the maximum batch size with respect to each fine-tuning method, instead of using a fixed one across different methods. Therefore, DP-BiTFiT can further leverage its memory efficiency to achieve the best throughput. Here we consider a setting of high-dimensional data ($T = 512^2$) but small ResNet ($11.7 \sim 58.2$M parameters) and the other setting of low-dimensional data ($T = 100$) but large GPT2 ($125 \sim 774$M parameters).

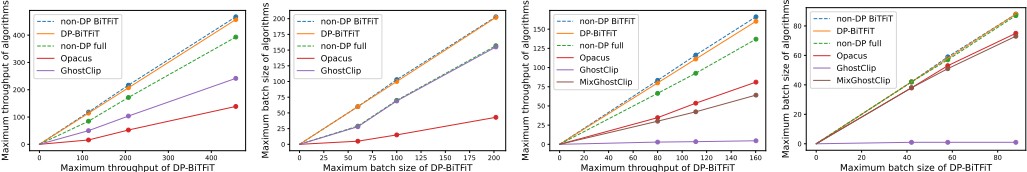

Figure 4: Maximum throughput and batch size by different fine-tuning methods. Left two: E2E dataset with GPT2-small/medium/large (MixGhostClip is equivalent to GhostClip for this small $T$). Right two: 50000 images of $512 \times 512$ pixels with ResNet 50/101/152.

## 4 EXPERIMENTS

We now test the accuracy of DP-BiTFiT on natural language and computer vision tasks, with the settings in Appendix D. For DP full fine-tuning algorithms, we use GhostClip (Li et al., 2021) on texts, and MixedGhostClip (Bu et al., 2022a) on images, which achieve SOTA efficiency and accuracy on these datasets respectively. We compute $\epsilon$ using a conversion from RDP though tighter privacy accountants in Section 2 are feasible. We illustrate in Table 17 that tuning the learning rate for BiTFiT is not difficult. And we observe in all experiments that, with or without DP, the optimal learning rate for BiTFiT is larger than that for full fine-tuning.

### 4.1 TEXT CLASSIFICATION

We experiment on MNLI-m(mismatch) (Williams et al., 2018), QQP (Iyer et al., 2017), QNLI (Rajpurkar et al., 2016), and SST2 datasets (Socher et al., 2013). Competitive algorithms include reparameterized gradient perturbation (RGP, (Yu et al., 2021c)), LoRA, Adapter and Compacter (Yu et al., 2021a). We use the same setup as (Li et al., 2021) on RoBERTa models, only increasing the

learning rate for DP-BiTFiT. Additional results with different clipping functions and under a stronger privacy guarantee $\epsilon = 3$ can be found in Table 12.

Table 3: Accuracy of fine-tuning methods with RoBERTa, under $\epsilon = 8$. More non-private fine-tuning results (similar to here) can be found in (Yu et al., 2021a; Hu et al., 2021; Zaken et al., 2022). Note that last-layer training of RoBERTa-base only gets 87.2% on SST2 and 77.3% on QNLI.

| | Full (Li et al., 2021) | | RGP (Yu et al., 2021a) | Adapter (Yu et al., 2021a) | LoRA (Yu et al., 2021a) | | BiTFiT Ours | | Compacter (Yu et al., 2021a) |
|---|---|---|---|---|---|---|---|---|---|
| Additional params to networks | ✗ | | ✗ | ✓ | ✓ | | ✗ | | ✓ |
| Forward caching activations | ✓ | | ✓ | ✓ | ✓ | | ✗ | | ✓ |
| *RoBERTa-base (125M)* | | | | | | | | | |
| % of trainable params | 100% | | 100% | 1.4% | 0.94% | | 0.083% | | 0.055% |
| | standard | DP | DP | DP | standard | DP | standard | DP | DP |
| Accuracy SST2 | 94.5 | 92.1 | 91.6 | 92.5 | 95.1 | 92.2 | 93.5 | 92.4 | 92.3 |
| Accuracy QNLI | 91.4 | 87.9 | 87.2 | 87.5 | 93.3 | 87.3 | 87.3 | 86.5 | 85.1 |
| Accuracy QQP | 87.3 | 86.1 | 85.5 | 85.6 | 90.8 | 85.7 | 86.1 | 83.4 | 84.7 |
| Accuracy MNLI-m | 85.9 | 83.2 | 80.1 | 83.4 | 87.5 | 83.5 | 83.4 | 82.6 | 82.6 |
| *RoBERTa-large (355M)* | | | | | | | | | |
| % of trainable params | 100% | | 100% | 1.4% | 0.94% | | 0.077% | | 0.053% |
| | standard | DP | DP | DP | standard | DP | standard | DP | DP |
| Accuracy SST2 | 96.2 | 93.8 | 93.0 | 93.9 | 96.2 | 95.3 | 95.5 | 94.5 | 94.2 |
| Accuracy QNLI | 93.6 | 91.1 | 90.0 | 90.7 | 94.9 | 90.8 | 92.2 | 91.0 | 90.2 |
| Accuracy QQP | 87.9 | 87.5 | 86.7 | 86.3 | 91.6 | 87.4 | 87.9 | 86.5 | 86.2 |
| Accuracy MNLI-m | 90.3 | 87.0 | 86.1 | 87.7 | 90.6 | 87.8 | 89.3 | 87.6 | 87.5 |

In Table 3, DP-BiTFiT is highly parameter efficiency and on-par with other DP fine-tuning in terms of accuracy. As indicated by Figure 1 and Figure 3, over $2\times$ speedup and over $3\times$ memory saving is observed, when switching from DP full fine-tuning to DP-BiTFiT across datasets.

*Remark* 4.1. It is encouraging to observe that the gap between the full fine-tuning and BiTFiT, with or without DP, tends to decrease as the model size increases. For instance on QNLI, this gap without privacy reduces from 4.1% to 1.4%, and with privacy reduces from 1.4% to 0.1%. This scaling pattern is consistently observed on different tasks, e.g. in Table 4 and Table 5.

## 4.2 NATURAL LANGUAGE GENERATION

We compare DP-BiTFiT with DP LoRA, full fine-tuning, and prefix tuning (Li & Liang, 2021) on E2E dataset (Dusek et al., 2020), in order to train GPT2 that generates texts to evaluate a restaurant. The performance measures are BLEU (Papineni et al., 2002), ROGUE-L (Lin, 2004), NIST (Sadjadi et al., 2018), METEOR (Banerjee & Lavie, 2005), CIDEr (Vedantam et al., 2015) and perplexity. We use the same setup as (Bu et al., 2022b) with automatic clipping, only increasing the learning rate for DP-BiTFiT. More results under a stronger privacy guarantee $\epsilon = 3$ can be found in Table 13.

Table 4: Performance of fine-tuning methods with GPT2, under $\epsilon = 8$. LoRA and prefix results are documented in (Li et al., 2021). Best performance in each model is in bold text.

| Model | Fine-tuning | % of params | Privacy↓ | Perplexity↓ | BLEU↑ | ROGUE-L↑ | NIST↑ | METEOR↑ | CIDEr↑ |
|---|---|---|---|---|---|---|---|---|---|
| GPT2-small (124M) | full | 100% | standard | 2.91 | 69.46 | 71.36 | 8.78 | 0.46 | 2.42 |
| | | | DP ($\epsilon = 8$) | 2.33 | **63.60** | 67.07 | **7.71** | 0.40 | 1.94 |
| | LoRA | — | standard | — | 69.68 | 71.71 | 8.82 | 0.46 | 2.49 |
| | | | DP ($\epsilon = 8$) | — | 63.39 | **67.53** | 7.45 | **0.41** | **1.95** |
| | prefix | — | standard | — | 68.85 | 70.81 | 8.72 | 0.45 | 2.35 |
| | | | DP ($\epsilon = 8$) | — | 49.26 | 60.73 | 5.53 | 0.36 | 1.57 |
| | BiTFiT | 0.082% | standard | 3.19 | 64.46 | 63.67 | 4.25 | 0.36 | 1.36 |
| | | | DP ($\epsilon = 8$) | 2.89 | 60.13 | 64.96 | 6.14 | 0.37 | 1.62 |
| GPT2-medium (355M) | full | 100% | standard | 2.08 | 68.50 | 71.46 | 8.63 | 0.45 | 2.14 |
| | | | DP ($\epsilon = 8$) | 2.25 | **64.22** | **67.53** | **8.17** | **0.42** | **2.08** |
| | BiTFiT | 0.076% | standard | 2.85 | 64.48 | 67.81 | 8.50 | 0.43 | 2.11 |
| | | | DP ($\epsilon = 8$) | 2.67 | 61.02 | 66.13 | 7.18 | 0.39 | 1.80 |
| GPT2-large (774M) | full | 100% | standard | 1.79 | 66.84 | 70.38 | 8.73 | 0.46 | 2.36 |
| | | | DP ($\epsilon = 8$) | 2.26 | 64.64 | **68.97** | 8.30 | **0.42** | **2.16** |
| | BiTFiT | 0.066% | standard | 2.79 | 65.79 | 67.61 | 8.55 | 0.43 | 2.21 |
| | | | DP ($\epsilon = 8$) | 2.59 | **65.21** | 67.88 | **8.43** | **0.42** | 2.15 |

In Table 4, DP-BiTFiT has shown strong performance, even outperforming DP full fine-tuning on GPT2-large, as well as both the computation and parameter efficiency (see Figure 4). Similar to Remark 4.1, the gap of BLEU score between DP-BiTFiT and DP full fine-tuning reduces from -3.06/-3.20 (GPT2-small/medium) to +0.57 (GPT2-large), as the model size increases. We refer to Table 13 for a more significant pattern when $\epsilon = 3$.

### 4.3 IMAGE CLASSIFICATION

We further experiment on CIFAR10/CIFAR100 ($32 \times 32$ pixels, resized to $224 \times 224$) and CelebA ($218 \times 178$ pixels, not resized) after pre-training on ImageNet ($224 \times 224$ pixels). For these downstream datasets (e.g. CIFAR10 has only 10 classes), the number of classes is different than that in ImageNet, which has 1000 classes. Consequently, the classification head of the pretrained model is re-placed by random initialization. Therefore, our DP-BiTFiT is applied on top of the last-layer training, but the number of trainable parameter remains $\approx 0.1\%$ of the model parameters. For instance, ViT-large has 303M parameters, of which 282k are biases and the weight of last layer contains $\approx 100$k, depending on the number of classes in the downstram task.

We observe that DP-BiTFiT enjoys $1.5\times$ speedup for transformers and ResNet in Table 16, and that DP-BiTFiT performs on par with full fine-tuning in Table 5,Table 14 and Table 15, e.g. achieving state-of-the-art 99.0% accuracy on CIFAR10 and 91.2% on CIFAR100 at $\epsilon = 2$. Our observation holds across various models (especially on transformers), privacy budgets, and datasets. However, DP-BiTFiT needs extra attention for convolutional neural networks (CNN) as we elaborate in Remark 4.2.
*Remark* 4.2. DP-BiTFiT may be less performant if the convolution layers do not contain biases, e.g. in many popular models including ResNet (He et al., 2016). This issue can be mitigated by enabling the biases in the model (not affecting non-DP performance) or warming up with full fine-tuning at early iterations. Leveraging these solutions empirically improves the DP accuracy of ResNet18 on CelebA [Smiling] classification from 88% to 92% (c.f. Appendix A.2 for detailed discussion).

Table 5: Accuracy of DP fine-tuning methods on CIFAR10 and CelebA. More results under different $\epsilon$ and network architectures can be found in Appendix E.3.

| Dataset | | Model | Fine-tuning | Accuracy |
|---|---|---|---|---|
| CIFAR10 ($\epsilon = 2, \delta =$1e-5) | (Yu et al., 2021b) | ResNet152 (GEP) | last-layer | 94.8 |
| | (Tramer & Boneh, 2020) | SIMCLRv2 | last-layer | 92.7 |
| | (De et al., 2022) | Wide-ResNet28 | last-layer | 93.6 |
| | | Wide-ResNet28 | full | 95.4 |
| | (Bu et al., 2022a) | crossvit-base-240 | full | 96.1 |
| | | vit-base-patch16 | full | 97.4 |
| | | vit-large-patch16 | full | 98.9 |
| | Ours | crossvit-base-240 | BiTFiT | 95.7 |
| | | vit-base-patch16 | BiTFiT | 97.7 |
| | | vit-large-patch16 | BiTFiT | 99.0 |
| CelebA [Smiling] ($\epsilon = 8, \delta =$5e-6) | (Bu et al., 2022b) | ResNet9 | full | 91.08 |
| | Ours | ResNet18 | full | 91.02 |
| | | ResNet18 | BiTFiT | 88.17 |
| | | ResNet18 | last-layer | 66.15 |
| CelebA [Male] ($\epsilon = 8, \delta =$5e-6) | (Bu et al., 2022b) | ResNet9 | full | 95.70 |
| | Ours | ResNet18 | full | 95.15 |
| | | ResNet18 | BiTFiT | 92.29 |
| | | ResNet18 | last-layer | 78.70 |
| CelebA [Multi-label] ($\epsilon = 8, \delta =$5e-6) | (Bu et al., 2022b) | ResNet9 | full | 87.58 |
| | Ours | ResNet18 | full | 88.38 |
| | | ResNet18 | BiTFiT | 86.87 |
| | | ResNet18 | last-layer | 83.67 |

Table 6: Accuracy of DP ViT-large on CIFAR10/100, 3 epochs, various $\epsilon$.

| CIFAR10 | DP last-layer | DP-BiTFiT | DP full |
|---|---|---|---|
| $\epsilon = 1$ | 98.4 | 98.9 | 98.9 |
| $\epsilon = 2$ | 98.6 | 99.0 | 98.9 |
| $\epsilon = 4$ | 98.6 | 99.0 | 99.0 |
| $\epsilon = 8$ | 98.7 | 99.0 | 99.0 |
| CIFAR100 | DP last-layer | DP-BiTFiT | DP full |
| $\epsilon = 1$ | 86.2 | 90.2 | 87.7 |
| $\epsilon = 2$ | 87.3 | 91.2 | 90.1 |
| $\epsilon = 4$ | 88.1 | 91.8 | 91.0 |
| $\epsilon = 8$ | 88.8 | 92.3 | 91.3 |

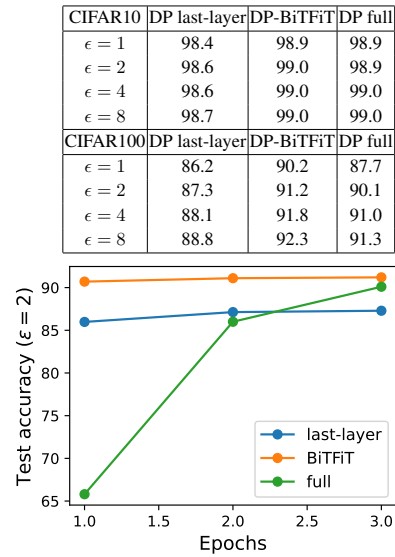

Figure 5: Accuracy of DP ViT-large on CIFAR100.

## 5 DISCUSSION

In this work, we study DP-BiTFiT to privately train the bias terms of neural networks. The highlight of DP-BiTFiT is the accuracy, the parameter efficiency and the computation efficiency, which is realized by not forward caching the activation tensors, and not back-propagating the gradient of weights. This consequently allows DP-BiTFiT to be as fast and memory-saving as its non-private counterpart, and thus particularly suitable for large models and high-dimension data, compared to full fine-tuning or other parameter-efficient methods.

For future directions, DP-BiTFiT can be readily combined with prefix-based tuning and *weights*-based fine-tuning, e.g. DP Adapter+BiTFiT and DP LoRA+BiTFiT, via $f(\boldsymbol{x}; \mathbf{W}_0, \hat{\mathbf{b}}, \hat{\boldsymbol{\theta}})$ using the notation in Equation (1). For instance, we can optimize only the embedding layer (which has no bias terms) and all bias terms in other layers. We expect this interpolating approach between full fine-tuning and BiTFiT to produce even better performance at greater efficiency.

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

# A   DETAILED ANALYSIS

## A.1   BACK-PROPAGATION

We rigorously analyze the neural network represented in Section 2: for sample index $i \in [B]$,

$$\underbrace{\boldsymbol{a}_{l+1,i}}_{\mathbb{R}^{T \times d'}} = \phi(\underbrace{\boldsymbol{s}_{l,i}}_{\mathbb{R}^{T \times p}}), \qquad \boldsymbol{s}_{l,i} = \underbrace{\boldsymbol{a}_{l,i}}_{\mathbb{R}^{T \times d}} \underbrace{\mathbf{W}_l}_{\mathbb{R}^{d \times p}} + \underbrace{\mathbf{1}}_{\mathbb{R}^{T \times 1}} \cdot \underbrace{\mathbf{b}_l}_{\mathbb{R}^{1 \times p}}, \tag{5}$$

Then the per-sample weight gradient is given by the chain rule as

$$\frac{\partial \mathcal{L}_i}{\partial \mathbf{W}_l}^\top = \sum_j \frac{\partial \mathcal{L}_i}{\partial \boldsymbol{s}_{l,j}}^\top \frac{\partial \boldsymbol{s}_{l,j}}{\partial \mathbf{W}_l} = \frac{\partial \mathcal{L}_i}{\partial \boldsymbol{s}_{l,i}}^\top \frac{\partial \boldsymbol{s}_{l,i}}{\partial \mathbf{W}_l} = \frac{\partial \mathcal{L}_i}{\partial \boldsymbol{s}_{l,i}}^\top \boldsymbol{a}_{l,i} = \frac{\partial \mathcal{L}}{\partial \boldsymbol{s}_{l,i}}^\top \boldsymbol{a}_{l,i}$$

in which the second equality holds when there is no parameter sharing (so that each per-sample loss only depends on $i$-th input and output). The last equality holds for the same reason.

Similarly, we have the per-sample bias gradient as

$$\frac{\partial \mathcal{L}_i}{\partial \mathbf{b}_l}^\top = \sum_j \frac{\partial \mathcal{L}_i}{\partial \boldsymbol{s}_{l,j}}^\top \frac{\partial \boldsymbol{s}_{l,j}}{\partial \mathbf{b}_l} = \frac{\partial \mathcal{L}_i}{\partial \boldsymbol{s}_{l,i}}^\top \frac{\partial \boldsymbol{s}_{l,i}}{\partial \mathbf{b}_l} = \frac{\partial \mathcal{L}_i}{\partial \boldsymbol{s}_{l,i}}^\top \mathbf{1} = \frac{\partial \mathcal{L}}{\partial \boldsymbol{s}_{l,i}}^\top \mathbf{1}.$$

We additionally demonstrate that bias gradient is independent of the input $\boldsymbol{a}_l$, on the convolution (1d/2d/3d) and the normalization layers. For the convolution, $\boldsymbol{s}_l$ is the inversely folded output and $\boldsymbol{a}_l$ is the unfolded input, then the forward pass is the same as that of linear layer in Equation (5). Notice that $T$ is the product of hidden feature dimension (c.f. Bu et al. (2022a)), which depends on the padding, kernel sizes, strides, etc. For the batch, layer, group, and instance normalization, the forward pass is

$$\boldsymbol{s}_{l,i} = \frac{\boldsymbol{a}_{l,i} - \mathbb{E}(\boldsymbol{a}_l)}{\sqrt{\mathrm{Var}(\boldsymbol{a}_l) + 0.00001}} \cdot \mathbf{W}_l + \mathbf{1} \cdot \mathbf{b}_l$$

which can be analyzed similarly to that of Equation (5).

## A.2   MAKING BiTFiT WORK WITH CONVOLUTIONAL NEURAL NETWORKS

Most (non-transformer) vision models use convolution layers and batch normalization during their standard non-DP training, which is problematic for DP training in general, especially for DP-BiTFiT. We take ResNet (He et al., 2016) as a concrete example.

Firstly, it is well-known that DP training does not support batch normalization, because the mean and standard deviation are computed based on samples (c.f. `https://opacus.ai/tutorials/guide_to_module_validator`). Therefore, in DP training, ResNet-BN (with batch normalization) is modified to a different achitecture ResNet-GN (replaced by group normalization, e.g. Abadi et al. (2016)). Put differently, ResNet is different in DP and non-DP training and sometimes the comparison may be unfair. This makes vision transformers favorable because they use layer normalization so that the architecures do not require modification when switching to DP regime.

Secondly, the convolution layers usually do not contain bias terms when followed by batch normalization. This is the case in packages like tensorflow.keras, torchvision, timm, and in models like ResNet, ResNext, DenseNet, etc. The reason of not having bias terms is that the batch normalization performs mean subtraction, which make the biases ineffective (see `https://discuss.pytorch.org/t/no-bias-in-the-pretrianed-state-dictionary-of-resnet18/153263/2`). In words, ResNet-BN(with bias)=ResNet-BN(no bias), but ResNet-GN(with bias)≠ResNet-GN(no bias).

**Consequences**   Consider two networks, ResNet(no bias) with bias-less convolution and ResNet(with bias). In full fine-tuning, we are training all 100 layers of both ResNets and they are equivalent under batch normalization; but in DP-BiTFiT, we are essentially not training ResNet(no bias), maybe except the classification head.

### A.2.1 WALK-AROUND 1

We can manually re-write the convolution layers in CNNs, which is technically troublesome and has to be done in a case-by-case manner. For example, in Bu et al. (2022b), ResNet9 was implemented with bias in the convolution layers. This walk-around can improve the performance of DP-BiTFiT significantly (because all layers are trainable now) without sacrificing the training efficiency.

### A.2.2 WALK-AROUND 2

Alternatively, we can leverage a two-phase training to interpolate between full fine-tuning and BiTFiT. We introduce the *two-phase training*, denoted as $X$+BiTFiT, which firstly applies DP full fine-tuning for $X$ epochs then DP-BiTFiT for the rest of training. Hence, $X$+BiTFiT becomes DP full fine-tuning when $X$ equals total epochs, and reduces to DP-BiTFiT when $X = 0$. Empirically speaking, it suffices to use $X \leq 2$ to achieve comparable accuracy to full fine-tuning, while still enjoying some speedup. The effectiveness of two-phase training is verified in Appendix E.3. 1+BiTFiT outperforms previous SOTA by DP full fine-tuning Bu et al. (2022a) that used BEiT-large: CIFAR10 $97.1\% \rightarrow 98.8\%$; CIFAR100 $86.2\% \rightarrow 88.7\%$, under $\epsilon = 2$. 2+BiTFiT is comparable to previous SOTA, $87.05/87.58\% \rightarrow 86.54/86.71\%$ on CelebA in Table 16, under $\epsilon = 3/8$ respectively.

## B  IMPLEMENTATION OF DP-BITFIT

In this section we describe the implementation of DP-BiTFiT, which only uses Pytorch backward hook but not the forward hook, and thus is different from existing packages such as FastGradClip Lee & Kifer (2020), Opacus Yousefpour et al. (2021), Private Transformers Li et al. (2021), Private CNN Bu et al. (2022a). Notice that in these packages, the forward hook is used to store the activation tensor $\boldsymbol{a}_l$ for all layers, which incurs huge memory burden as discussed in Section 2.

The Pytorch backward hook is a function, to be registered on a torch Module (or a layer in the neural network), that will be executed in the backward propagation. The backward hook automatically extracts the input gradient $\frac{\partial \mathcal{L}}{\partial \boldsymbol{a}_l}$ and the output gradient $\frac{\partial \mathcal{L}}{\partial \boldsymbol{s}_l}$ of the layer.

In DP-BiTFiT, we call `register_backward_hook` to register a backward hook for Line 5 of Algorithm 1. An example for a linear layer: $\mathbb{R}^{B \times T \times d} \rightarrow \mathbb{R}^{B \times T \times p}$ looks like

```
def hook(linear_layer, grad_input, grad_output):
    linear_layer.bias.grad_sample = grad_output.sum(dim=1)
    linear_layer.bias.norm_sample = linear_layer.bias.grad_sample.norm(2,dim=1)
```

Here the attribute `norm_sample` stores the per-sample gradient norm $\left\|\frac{\partial \mathcal{L}_i}{\partial \mathbf{b}_l}\right\|_F$, and the attribute `grad_sample` stores the $\mathbb{R}^{B \times p}$ per-sample gradient of bias.

Then the implementation of DP-BiTFiT for one iteration looks like

```
output=model(input)
loss=F.cross_entropy()(output,label)
torch.autograd.grad(loss,biases)
all_layer_norm_sample = torch.stack([param.norm_sample for param in biases],dim=0).norm(2, dim=0)
clipping_factor=1/(all_layer_norm_sample+0.01)
for layer in model.modules():
    layer.bias.grad=torch.einsum("i,i...->...", clipping_factor,layer.bias.grad_sample)
optimizer.step()
optimizer.zero_grad()
```

where `biases` is the collection of all bias terms in all layers.

## C  COMPLEXITY ANALYSIS

We provide more details on analyzing the time and space complexity. The analysis for full fine-tuning has been presented in Appendix C of Bu et al. (2022a) and is adapted here for the parameter efficient fine-tuning: for example, Adapter Houlsby et al. (2019) uses two matrices $W_{down} \in \mathbb{R}^{p \times r}, W_{up} \in \mathbb{R}^{r \times p}$ that constitute

$$x \longleftarrow x + \text{GeLU}(x \cdot W_{down})W_{up}$$

Hence the complexity, in comparison to full-finetuning, changes by replacing $d \to 2r$.

LoRA Hu et al. (2021) also uses two matrices $W_{down} \in \mathbb{R}^{d \times r}, W_{up} \in \mathbb{R}^{r \times p}$ that constitute

$$x \longleftarrow x \cdot W + x \cdot W_{down} W_{up}$$

Hence the complexity, in comparison to full-finetuning, changes by replacing $pd \to r(p + d)$.

Table 7: Per-layer time and space complexity of training on weights (full and parameter efficient fine-tuning) and biases. '+' means additional overhead to non-DP training.

| | forward &output grad | weight training | | | | bias training | |
|---|---|---|---|---|---|---|---|
| | | non-DP | DP full (Opacus) | DP LoRA | DP Adapter | non-DP | DP (ours) |
| Time complexity | $4BTpd$ | $2BTpd$ | $+2BTpd$ | $+2BT(pr + dr)$ | $+4BTpr$ | $BTp$ | $+3Bp$ |
| Space complexity | $pd + BTd$ | $BT(p + d)$ | $+Bpd$ | $+B(pr + dr)$ | $+2Bpr$ | $p$ | $+Bp$ |
| # back-prop | | 1 | 1 | 1 | 1 | 1 | 1 |
| forward hook | | ✗ | ✓ | ✓ | ✓ | ✗ | ✗ |

For per-sample bias gradient clipping, we need ${\frac{\partial \mathcal{L}_i}{\partial \mathbf{b}_l}}^\top = {\frac{\partial \mathcal{L}}{\partial \boldsymbol{s}_{l,i}}}^\top \mathbf{1}$ in Equation (4), which consists of the *per-sample gradient instantiation* (i.e. summation along the feature dimension, from $\mathbb{R}^{Tp} \to \mathbb{R}^p$, $\frac{\partial \mathcal{L}}{\partial \boldsymbol{s}_{l,i}} \to \frac{\partial \mathcal{L}_i}{\partial \mathbf{b}_l}$), and computing the per-sample gradient norm (i.e. *taking the square* at each index and *summing all indices*). Here each operation in italic takes $Bp$ time complexity, meaning the total time complexity is $3Bp$, but the space complexity is $Bp$ if operated in-place.

## D  EXPERIMENT DETAILS

### D.1  LANGUAGE TASKS

Throughout this work, the text datasets are processed and loaded from Huggingface Lhoest et al. (2021). We follow the same setup as Li et al. (2021); Bu et al. (2022b), such as $\delta = 0.5/\text{sample size}$. The full fine-tuning is implemented by Private Transformers codebase, version 0.2.0 (i.e. GhostClip algorithm Li et al. (2021)).

For text classification, we experiment on four datasets: **MNLI(m)**, the matched splits from Multi-Genre Natural Language Inference Corpus; **QQP**, the Quora Question Pairs2 dataset; **QNLI** The Stanford Question Answering dataset; **SST2** The Stanford Sentiment Treebank dataset.

To give a fair comparison, we use the same optimizer as in Li et al. (2021), i.e. DP-Adam with Abadi's clipping.

Table 8: Hyperparameters of text classification in Table 3 and Table 12, using RoBERTa (base/large).

| Dataset | MNLI | QQP | QNLI | SST2 |
|---|---|---|---|---|
| epoch | 18 | 18 | 6 | 3 |
| batch size | 6000 | 6000 | 2000 | 1000 |
| clipping threshold $R$ | 0.1 | | | |
| DP learning rate | full 5e-4 / BiTFiT 5e-3 | | | |
| non-DP learning rate | full 5e-5 / BiTFiT 1e-3 | | | |
| max sequence length | 256 | | | |

For E2E generation task, we experiment GPT2 models using the same optimizer as in Bu et al. (2022b), using DP-AdamW with automatic clipping.

### D.2  IMAGE TASKS

We give the experiments settings for image classification. For CIFAR10 and CIFAR100, we use the same setting as Bu et al. (2022a), e.g. 5 epochs for CrossViT, 3 epochs for ViT and BEiT-large. For CelebA, we use the same setting as Bu et al. (2022b), e.g. 10 epochs.

Table 9: Hyperparameters of E2E generation task in Table 4 and Table 13, using GPT2.

| Model | GPT2-small | GPT2-medium | GPT2-large |
|---|---|---|---|
| epoch | | 10 | |
| batch size | | 1024 | |
| DP learning rate (full) | 2e-3 | 2e-3 | 2e-3 |
| non-DP learning rate (full) | 2e-4 | 1e-4 | 1e-4 |
| DP learning rate (BiTFiT) | | 1e-2 | |
| non-DP learning rate (BiTFiT) | | 2e-3 | |
| learning rate decay | | No | |
| max sequence length | | 100 | |

We use DP-Adam with Abadi's clipping. We do not apply tricks such as random data augmentation, weight standardization Qiao et al. (2019), or parameter averaging Polyak & Juditsky (1992). Our experiments are heavily based on Private CNN (i.e. MixGhostClip algorithm Bu et al. (2022a)) and TIMM codebases.

Table 10: Hyperparameters of image classification task in Section 4.3,Table 14,Table 15,Table 16.

| Dataset | CIFAR10 | CIFAR10 | CIFAR100 | CelebA |
|---|---|---|---|---|
| Model | CrossViT | ViT-large | ViT-large | ResNet18 |
| epoch | 5 | 3 | 3 | 10 |
| batch size | 1000 | 1000 | 1000 | 500 |
| clipping threshold | | 0.1 | | |
| DP learning rate (full) | 1e-3 | 5e-4 | 5e-4 | 1e-3 |
| DP learning rate (BiTFiT) | 5e-3 | 5e-3 | 5e-3 | 8e-3 |
| learning rate decay | | No | | |
| normalizing data | Yes | Yes | Yes | No |

# E  ADDITIONAL TABLES AND FIGURES

## E.1  PARAMETER EFFICIENCY OF DP-BITFIT

Table 11: Parameter efficiency of (DP) BiTFiT on various models.

| Model | Number of params | % of params |
|---|---|---|
| VGG11 | 133M | 0.009 |
| VGG16 | 138M | 0.009 |
| VGG19 | 144M | 0.010 |
| ResNet18 | 11.7M | 0.043 |
| ResNet34 | 21.8M | 0.044 |
| ResNet50 | 25.6M | 0.113 |
| ResNet101 | 44.5M | 0.121 |
| ResNet152 | 60.2M | 0.127 |
| wide_resnet50_2 | 68.9M | 0.051 |
| wide_resnet101_2 | 126.9M | 0.055 |
| convnext_base | 88.6M | 0.148 |
| convnext_large | 197.8M | 0.099 |
| ViT-small-patch16 | 22.0M | 0.238 |
| ViT-base-patch16 | 86.6M | 0.120 |
| ViT-large-patch16 | 304M | 0.090 |
| beit_base_patch16_224 | 86.5M | 0.088 |
| deit_base_patch16_224 | 86.4M | 0.120 |
| GPT2-small | 124M | 0.082 |
| GPT2-medium | 355M | 0.076 |
| GPT2-large | 774M | 0.066 |
| RoBERTa-base | 125M | 0.083 |
| RoBERTa-large | 355M | 0.077 |
| BERT-base-uncased | 109M | 0.094 |
| BERT-large-uncased | 335M | 0.081 |
| BART-large | 406M | 0.082 |
| longformer-base-4096 | 149M | 0.088 |
| longformer-large-4096 | 435M | 0.080 |

## E.2  MORE RESULTS ON DP-BITFIT AND LANGUAGE TASKS

Table 12: Accuracy of full fine-tuning and BiTFiT with RoBERTa, under different per-sample clipping functions (indicated as subscript, Abadi Abadi et al. (2016) and AUTO-S Bu et al. (2022b)). Same setting as Appendix D.

| | full (Li et al., 2021; Bu et al., 2022b) | | | | | BiTFiT (ours) | | | | |
|---|---|---|---|---|---|---|---|---|---|---|
| | \multicolumn RoBERTa-base | | | | | | | | | |
| | standard | $DP_{Abadi}$ | $DP_{AUTO}$ | $DP_{Abadi}$ | $DP_{AUTO}$ | standard | $DP_{Abadi}$ | $DP_{AUTO}$ | $DP_{Abadi}$ | $DP_{AUTO}$ |
| | $\epsilon=\infty$ | $\epsilon=8$ | $\epsilon=8$ | $\epsilon=3$ | $\epsilon=3$ | $\epsilon=\infty$ | $\epsilon=8$ | $\epsilon=8$ | $\epsilon=3$ | $\epsilon=3$ |
| Accuracy SST2 | 94.5 | 92.1 | 92.4 | 91.9 | 92.3 | 93.5 | 92.4 | 92.4 | 92.0 | 92.0 |
| Accuracy QNLI | 91.4 | 87.9 | 87.9 | 87.4 | 86.9 | 87.3 | 86.5 | 86.7 | 86.4 | 86.1 |
| Accuracy QQP | 87.3 | 86.1 | 86.6 | 85.6 | 85.8 | 86.1 | 83.4 | 84.0 | 83.0 | 83.8 |
| Accuracy MNLI-m | 85.9 | 83.2 | 83.8 | 82.5 | 83.2 | 83.4 | 82.6 | 82.6 | 81.5 | 82.0 |
| | \multicolumn RoBERTa-large | | | | | | | | | |
| | standard | $DP_{Abadi}$ | $DP_{AUTO}$ | $DP_{Abadi}$ | $DP_{AUTO}$ | standard | $DP_{Abadi}$ | $DP_{AUTO}$ | $DP_{Abadi}$ | $DP_{AUTO}$ |
| | $\epsilon=\infty$ | $\epsilon=8$ | $\epsilon=8$ | $\epsilon=3$ | $\epsilon=3$ | $\epsilon=\infty$ | $\epsilon=8$ | $\epsilon=8$ | $\epsilon=3$ | $\epsilon=3$ |
| Accuracy SST2 | 96.2 | 93.8 | 94.6 | 93.0 | 93.9 | 95.5 | 94.5 | 94.7 | 94.5 | 94.6 |
| Accuracy QNLI | 93.6 | 91.1 | 91.5 | 90.8 | 91.0 | 92.2 | 91.0 | 91.1 | 90.3 | 90.8 |
| Accuracy QQP | 87.9 | 86.9 | 87.5 | 86.6 | 86.8 | 87.9 | 86.5 | 87.1 | 86.3 | 86.5 |
| Accuracy MNLI-m | 90.3 | 87.0 | 87.1 | 86.4 | 86.3 | 89.3 | 87.6 | 87.7 | 87.2 | 87.2 |

Table 13: Accuracy of fine-tuning with GPT2 on E2E dataset. LoRA and prefix results are taken from Li et al. (2021). Same setting as Appendix D.

| Model | Fine-tuning | % of params | Privacy↓ | Perplexity↓ | BLEU↑ | ROGUE-L↑ | NIST↑ | METEOR↑ | CIDEr↑ |
|---|---|---|---|---|---|---|---|---|---|
| GPT2-small (124M) | full | 100% | standard | 2.91 | 69.46 | 71.36 | 8.78 | 0.46 | 2.42 |
| | | | DP ($\epsilon = 8$) | 2.33 | 63.60 | 67.07 | 7.71 | 0.40 | 1.94 |
| | | | DP ($\epsilon = 3$) | 2.36 | 61.34 | 65.87 | 7.07 | 0.39 | 1.80 |
| | LoRA | — | standard | — | 69.68 | 71.71 | 8.82 | 0.46 | 2.49 |
| | | | DP ($\epsilon = 8$) | — | 63.39 | 67.53 | 7.45 | 0.41 | 1.95 |
| | | | DP ($\epsilon = 3$) | — | 58.15 | 65.77 | 5.46 | 0.37 | 1.58 |
| | prefix | — | standard | — | 68.85 | 70.81 | 8.72 | 0.45 | 2.35 |
| | | | DP ($\epsilon = 8$) | — | 49.26 | 60.73 | 5.53 | 0.36 | 1.57 |
| | | | DP ($\epsilon = 3$) | — | 47.77 | 58.96 | 5.25 | 0.36 | 1.51 |
| | BiTFiT | 0.082% | standard | 3.19 | 64.46 | 63.67 | 4.25 | 0.36 | 1.36 |
| | | | DP ($\epsilon = 8$) | 2.89 | 60.13 | 64.96 | 6.14 | 0.37 | 1.62 |
| | | | DP ($\epsilon = 3$) | 3.00 | 54.78 | 63.55 | 4.78 | 0.34 | 1.31 |
| GPT2-medium (355M) | full | 100% | standard | 2.08 | 68.50 | 71.46 | 8.63 | 0.45 | 2.14 |
| | | | DP ($\epsilon = 8$) | 2.25 | 64.22 | 67.53 | 8.17 | 0.42 | 2.08 |
| | | | DP ($\epsilon = 3$) | 2.62 | 63.85 | 67.07 | 7.11 | 0.39 | 1.75 |
| | BiTFiT | 0.076% | standard | 2.85 | 64.48 | 67.81 | 8.50 | 0.43 | 2.11 |
| | | | DP ($\epsilon = 8$) | 2.67 | 61.02 | 66.13 | 7.18 | 0.39 | 1.80 |
| | | | DP ($\epsilon = 3$) | 2.67 | 57.11 | 66.16 | 5.07 | 0.37 | 1.47 |
| GPT2-large (774M) | full | 100% | standard | 1.79 | 66.84 | 70.38 | 8.73 | 0.46 | 2.36 |
| | | | DP ($\epsilon = 8$) | 2.26 | 64.64 | 68.97 | 8.30 | 0.42 | 2.16 |
| | | | DP ($\epsilon = 3$) | 2.65 | 64.18 | 67.86 | 7.94 | 0.40 | 2.01 |
| | BiTFiT | 0.066% | standard | 2.79 | 65.79 | 67.61 | 8.55 | 0.43 | 2.21 |
| | | | DP ($\epsilon = 8$) | 2.59 | 65.21 | 67.88 | 8.43 | 0.42 | 2.15 |
| | | | DP ($\epsilon = 3$) | 2.61 | 65.18 | 67.90 | 8.34 | 0.42 | 2.12 |

### E.3 MORE RESULTS ON TWO-PHASE TRAINING

Here X+BiTFiT does not train last layer, i.e. the classification head is randomized before full fine-tuning happens.

Table 14: Accuracy of two-phase fine-tuning on CIFAR10. Same setting as Appendix D.2. BEiT-large uses DP full fine-tuning learning rate 5e-4, DP-BiTFiT learning rate 5e-3. Others use DP full fine-tuning learning rate 1e-3, DP-BiTFiT learning rate 5e-3.

| CIFAR10 | | | | | |
|---|---|---|---|---|---|
| Model | Privacy | 0+BiTFiT | 1+BiTFiT | 2+BiTFiT | DP full |
| beit_large_patch16_224 | $\epsilon = 1$ | 11.7 | 98.2 | 97.9 | 97.2 |
| | $\epsilon = 2$ | 10.0 | 98.3 | 98.0 | 97.3 |
| | $\epsilon = 4$ | 13.8 | 98.2 | 98.0 | 97.5 |
| | $\epsilon = 8$ | 10.1 | 98.5 | 98.0 | 97.8 |
| beit_base_patch16_224 | $\epsilon = 1$ | 10.0 | 96.6 | 96.0 | 95.4 |
| | $\epsilon = 2$ | 10.7 | 97.1 | 96.4 | 96.0 |
| | $\epsilon = 4$ | 14.0 | 97.2 | 96.6 | 96.2 |
| | $\epsilon = 8$ | 10.0 | 97.2 | 96.5 | 96.3 |
| deit_base_patch16_224 | $\epsilon = 1$ | 78.2 | 94.4 | 95.2 | 95.4 |
| | $\epsilon = 2$ | 75.0 | 95.4 | 95.2 | 95.6 |
| | $\epsilon = 4$ | 72.9 | 95.8 | 95.9 | 96.0 |
| | $\epsilon = 8$ | 71.2 | 96.1 | 96.0 | 96.3 |
| crossvit_base_240 | $\epsilon = 1$ | 74.3 | 92.4 | 94.3 | 95.2 |
| | $\epsilon = 2$ | 80.4 | 93.6 | 95.0 | 95.3 |
| | $\epsilon = 4$ | 81.0 | 94.9 | 95.8 | 95.7 |
| | $\epsilon = 8$ | 78.2 | 94.8 | 95.8 | 96.2 |
| vit_large_patch16_224 | $\epsilon = 1$ | 89.7 | 98.9 | 98.7 | 98.9 |
| | $\epsilon = 2$ | 90.6 | 98.8 | 98.9 | 98.9 |
| | $\epsilon = 4$ | 93.2 | 98.9 | 98.8 | 99.0 |
| | $\epsilon = 8$ | 93.9 | 99.0 | 98.9 | 99.0 |
| vit_base_patch16_224 | $\epsilon = 1$ | 86.7 | 95.2 | 97.0 | 96.8 |
| | $\epsilon = 2$ | 89.3 | 97.7 | 97.1 | 97.1 |
| | $\epsilon = 4$ | 88.3 | 97.7 | 97.2 | 97.2 |
| | $\epsilon = 8$ | 88.7 | 97.6 | 97.2 | 97.4 |

Table 15: Accuracy of two-phase fine-tuning on CIFAR100. Same setting as Appendix D.2. BEiT-large uses DP full fine-tuning learning rate 5e-4, DP-BiTFiT learning rate 5e-3. Others use DP full fine-tuning learning rate 1e-3, DP-BiTFiT learning rate 5e-3.

| CIFAR100 | | | | | |
|---|---|---|---|---|---|
| Model | Privacy | 0+BiTFiT | 1+BiTFiT | 2+BiTFiT | DP full |
| beit_large_patch16_224 | $\epsilon = 1$ | 1.0 | 86.9 | 87.8 | 87.0 |
| | $\epsilon = 2$ | 1.0 | 88.7 | 89.3 | 88.7 |
| | $\epsilon = 4$ | 1.0 | 89.7 | 89.7 | 89.6 |
| | $\epsilon = 8$ | 1.0 | 90.3 | 90.7 | 90.0 |
| beit_base_patch16_224 | $\epsilon = 1$ | 1.0 | 81.4 | 82.2 | 80.9 |
| | $\epsilon = 2$ | 1.0 | 83.4 | 83.4 | 83.1 |
| | $\epsilon = 4$ | 1.0 | 84.6 | 85.1 | 84.8 |
| | $\epsilon = 8$ | 1.0 | 84.9 | 85.6 | 85.2 |
| deit_base_patch16_224 | $\epsilon = 1$ | 10.9 | 49.1 | 65.9 | 69.1 |
| | $\epsilon = 2$ | 13.6 | 58.1 | 71.5 | 74.3 |
| | $\epsilon = 4$ | 15.7 | 64.5 | 73.9 | 77.1 |
| | $\epsilon = 8$ | 16.6 | 69.7 | 75.7 | 77.9 |
| crossvit_base_240 | $\epsilon = 1$ | 12.2 | 49.2 | 61.7 | 67.6 |
| | $\epsilon = 2$ | 12.3 | 56.8 | 65.3 | 71.6 |
| | $\epsilon = 4$ | 17.2 | 61.6 | 70.4 | 73.1 |
| | $\epsilon = 8$ | 20.9 | 63.4 | 72.8 | 74.2 |
| vit_large_patch16_224 | $\epsilon = 1$ | 14.0 | 73.5 | 86.0 | 87.7 |
| | $\epsilon = 2$ | 19.4 | 82.4 | 89.0 | 90.1 |
| | $\epsilon = 4$ | 24.3 | 87.5 | 89.9 | 91.0 |
| | $\epsilon = 8$ | 23.9 | 89.0 | 90.7 | 91.3 |
| vit_base_patch16_224 | $\epsilon = 1$ | 16.0 | 64.3 | 79.5 | 83.9 |
| | $\epsilon = 2$ | 22.9 | 77.0 | 83.8 | 85.5 |
| | $\epsilon = 4$ | 21.2 | 83.0 | 85.2 | 87.2 |
| | $\epsilon = 8$ | 26.2 | 83.8 | 86.5 | 87.1 |

## F    EXTRA

Table 16: Accuracy on CelebA dataset with settings in Appendix D.2 from one run. DP full fine-tuning is implemented with the most efficient MixGhostClip algorithm Bu et al. (2022a). We observe that linear probing (LP) only gives 83.67% at $\epsilon = 8$. *Note the accuracy is based on `timm<=0.6.5` and may change for a different version.

| Attributes | 0+BiTFiT | 1+BiTFiT | 2+BiTFiT | DP full | DP-BiTFiT(LP) | 0+BiTFiT | 1+BiTFiT | 2+BiTFiT | DP full | DP-BiTFiT(LP) |
|---|---|---|---|---|---|---|---|---|---|---|
| | | | $\epsilon = 3$ | | | | | $\epsilon = 8$ | | |
| 5 o Clock Shadow | 90.01 | 90.01 | 90.14 | 91.32 | 90.35 | 90.01 | 90.01 | 90.51 | 91.64 | 90.97 |
| Arched Eyebrows | 71.56 | 73.12 | 76.01 | 77.33 | 75.41 | 71.56 | 73.74 | 75.49 | 78.82 | 76.49 |
| Attractive | 68.71 | 73.98 | 75.99 | 79.22 | 74.96 | 69.70 | 73.61 | 76.20 | 78.08 | 7523 |
| Bags Under Eyes | 79.74 | 79.76 | 81.27 | 81.73 | 81.14 | 79.74 | 79.74 | 80.69 | 82.62 | 8172 |
| Bald | 97.88 | 97.88 | 97.88 | 97.93 | 97.93 | 97.88 | 97.88 | 97.88 | 97.91 | 9790 |
| Bangs | 84.43 | 84.43 | 84.80 | 94.06 | 90.85 | 84.43 | 84.44 | 86.51 | 94.22 | 92.34 |
| Big Lips | 67.30 | 67.30 | 67.30 | 67.78 | 67.42 | 67.30 | 67.30 | 67.29 | 68.34 | 67.65 |
| Big Nose | 78.80 | 78.95 | 80.08 | 81.19 | 79.96 | 78.80 | 78.92 | 79.23 | 81.86 | 80.28 |
| Black Hair | 72.84 | 74.86 | 82.37 | 85.84 | 81.48 | 73.02 | 78.71 | 83.33 | 86.47 | 82.38 |
| Blond Hair | 89.54 | 93.00 | 93.28 | 94.17 | 93.03 | 89.13 | 92.62 | 93.88 | 94.34 | 93.51 |
| Blurry | 94.94 | 94.94 | 94.94 | 95.05 | 95.21 | 94.94 | 94.94 | 94.96 | 95.10 | 95.34 |
| Brown Hair | 82.03 | 82.02 | 82.87 | 85.44 | 82.68 | 82.03 | 82.37 | 83.49 | 85.04 | 82.88 |
| Bushy Eyebrows | 87.05 | 87.05 | 87.21 | 88.26 | 87.11 | 87.05 | 87.05 | 87.15 | 89.02 | 87.22 |
| Chubby | 94.70 | 94.70 | 94.70 | 94.84 | 94.57 | 94.70 | 94.70 | 94.70 | 94.78 | 94.47 |
| Double Chin | 95.43 | 95.43 | 95.43 | 95.49 | 95.34 | 95.43 | 95.43 | 95.43 | 95.39 | 95.26 |
| Eyeglasses | 93.54 | 93.54 | 93.54 | 94.30 | 94.77 | 93.54 | 93.54 | 93.54 | 95.85 | 96.32 |
| Goatee | 95.42 | 95.42 | 95.42 | 95.96 | 95.41 | 95.42 | 95.42 | 95.42 | 95.89 | 95.55 |
| Gray Hair | 96.81 | 96.81 | 96.85 | 97.44 | 96.78 | 96.81 | 96.81 | 97.12 | 97.45 | 96.59 |
| Heavy Makeup | 76.51 | 82.76 | 85.71 | 88.48 | 83.73 | 77.22 | 83.03 | 85.86 | 89.05 | 84.70 |
| High Cheekbones | 62.13 | 68.20 | 81.63 | 83.77 | 76.91 | 61.43 | 67.27 | 81.33 | 84.20 | 79.42 |
| Male | 80.37 | 88.47 | 91.52 | 94.73 | 89.92 | 82.04 | 88.52 | 92.14 | 95.19 | 90.69 |
| Mouth Slightly Open | 54.03 | 59.32 | 77.61 | 86.75 | 74.20 | 55.26 | 60.70 | 79.42 | 90.24 | 77.53 |
| Mustache | 96.13 | 96.13 | 96.13 | 96.10 | 96.06 | 96.13 | 96.13 | 96.13 | 96.12 | 95.98 |
| Narrow Eyes | 85.13 | 85.13 | 85.13 | 85.14 | 85.15 | 85.13 | 85.13 | 85.13 | 85.16 | 85.13 |
| No Beard | 85.37 | 85.87 | 87.56 | 92.94 | 88.33 | 85.37 | 85.88 | 88.59 | 93.59 | 89.81 |
| Oval Face | 70.44 | 70.94 | 71.50 | 73.11 | 71.51 | 70.44 | 71.48 | 71.92 | 71.77 | 71.25 |
| Pale Skin | 95.79 | 95.79 | 95.79 | 95.79 | 95.76 | 95.79 | 95.79 | 95.79 | 95.79 | 95.73 |
| Pointy Nose | 71.43 | 71.51 | 71.63 | 71.89 | 71.40 | 71.43 | 71.47 | 71.77 | 72.87 | 72.11 |
| Receding Hairline | 91.51 | 91.51 | 91.51 | 91.59 | 91.40 | 91.51 | 91.51 | 91.51 | 91.61 | 91.39 |
| Rosy Cheeks | 92.83 | 92.83 | 92.86 | 93.07 | 92.75 | 92.87 | 92.83 | 92.86 | 93.33 | 92.99 |
| Sideburns | 95.36 | 95.36 | 95.36 | 96.44 | 95.55 | 95.36 | 95.36 | 95.36 | 96.63 | 95.79 |
| Smiling | 60.07 | 66.32 | 85.85 | 89.34 | 79.99 | 58.92 | 65.97 | 85.55 | 89.11 | 82.82 |
| Straight Hair | 79.01 | 79.01 | 79.02 | 79.65 | 79.22 | 79.01 | 79.01 | 79.13 | 78.60 | 79.47 |
| Wavy Hair | 71.24 | 73.09 | 76.22 | 77.35 | 77.98 | 70.86 | 73.62 | 77.11 | 72.73 | 78.90 |
| Wearing Earrings | 79.34 | 79.34 | 80.37 | 83.24 | 81.54 | 79.34 | 79.34 | 80.71 | 84.36 | 82.65 |
| Wearing Hat | 95.80 | 95.80 | 95.80 | 96.01 | 95.95 | 95.80 | 95.80 | 95.80 | 97.02 | 96.63 |
| Wearing Lipstick | 80.61 | 87.90 | 89.81 | 91.59 | 87.54 | 80.35 | 87.20 | 89.56 | 91.94 | 88.16 |
| Wearing Necklace | 86.21 | 86.21 | 86.21 | 86.21 | 86.16 | 86.21 | 86.21 | 86.21 | 86.21 | 86.12 |
| Wearing Necktie | 92.99 | 92.99 | 93.03 | 93.58 | 93.61 | 92.99 | 92.99 | 93.11 | 93.57 | 94.13 |
| Young | 75.71 | 79.33 | 81.23 | 83.69 | 80.57 | 75.71 | 78.52 | 80.66 | 83.11 | 80.93 |
| Average | 82.97 | 84.42 | 86.54 | 88.20 | 86.25 | 83.01 | 84.52 | 86.71 | 88.38 | 86.87 |
| Total time | 10:30 | 12:02 | 13:34 | 25:50 | 10:30 | 10:30 | 12:02 | 13:34 | 25:50 | 10:30 |

Table 17: Test accuracy on SST2 under $\epsilon = 8$, using DP-Adam with AUTO-S clipping.

| learning rate | DP-BiTFiT | | | | | DP full | | | | non-DP full | | | |
|---|---|---|---|---|---|---|---|---|---|---|---|---|---|
| | 5e-4 | 1e-3 | 2e-3 | 5e-3 | 1e-2 | 1e-4 | 2e-4 | 5e-4 | 1e-3 | 1e-5 | 2e-5 | 5e-5 | 1e-4 |
| RoBERTa-base | 90.94 | 91.28 | 91.74 | 92.43 | 90.94 | 91.51 | 91.97 | 92.43 | 91.28 | 93.92 | 94.38 | 94.49 | 93.35 |
| RoBERTa-large | 94.38 | 95.07 | 94.38 | 94.50 | 94.04 | 94.84 | 94.72 | 94.61 | 92.66 | 95.76 | 96.21 | 96.21 | 95.99 |

