# OpenReview forum: "Differentially Private Bias-Term Fine-tuning of Foundation Models"
_ICLR.cc/2024/Conference — Submitted to ICLR 2024_

### Official Review · Reviewer_tcjD · 2023-10-24

**Soundness:** 3 good
**Presentation:** 2 fair
**Contribution:** 2 fair
**Rating:** 5
**Confidence:** 3

**Summary:**

The paper describes how to perform BitFit parameter-efficient fine-tuning of bias terms only with the addition of noise for differential privacy. BitFit is known to be effective and very parameter efficient, and it works well with DP because the number of trained parameters is so low.

**Strengths:**

A strong case is made for using the combination of BitFit and DP-SGD. The memory and time savings are efficient, both in theory and according to the extensive experiments. It feels somewhat incremental, but the experimental contribution demonstrating that the comination works well is still significant.

**Weaknesses:**

Contrary to the claims in the paragraph "novelty", it seems to me there is not a huge creative leap in putting together BitFit and DP-SGD as is done here. Is there something I am missing? What is the "substantial algorithmic innovation"? What is the naive way of combining these ideas that your algorithm improves upon?

The writing is unclear in places:
* The authors should study how to use \citep and \citet appropriately.
* Why is $C_i$ given as an input to Algorithm 1? I wonder if you mean the clipping function $C$.

I think this work is useful, and should be shared at a workshop maybe, but unless my concerns about novelty are satisfied I'm not sure it merits publishing at a top-tier conference.

**Questions:**

* See question about novelty above.
* Is this any better from a clipping perspective? Do you have to materialize the per-example bias gradients in order to compute their norms? You could combine this with ghost clipping, right?

---

> ### Author Response · Authors · 2023-11-21
>
> We thank the reviewer for the comment. Here is a point-to-point response.
>
> 1. Contrary to the claims in the paragraph "novelty", it seems to me there is not a huge creative leap in putting together BitFit and DP-SGD as is done here. What is the "substantial algorithmic innovation"? What is the naive way of combining these ideas that your algorithm improves upon?
>
> **Response** We agree that BiTFiT and DP-SGD are both existing methods. However, our main contributions also include the complexity analysis and engineering implementation. 1) Complexity analysis of DP parameter-efficient fine-tuning (PEFT): DP-BiTFiT in main text, DP-LoRA and so on in appendix C. This is a missing piece in previous DP PEFT literature and significantly helpful in determining the benefit of applying different PEFT methods. E.g. we rigorously show the complexity saving is 50%, and the benefit of BiTFiT on long-context language tasks or high-resolution image tasks, i.e. the computation overhead of DP-BiTFiT in the last column of Table 2 is independent of T. This analysis is also missing in the non-DP BiTFiT. Without this analysis, our understanding of efficient fine-tuning methods will be at most empirical, and certain special properties of DP-BiTFiT (like activation-free optimization and feature-dimension-free overhead) will not be revealed. 2) Engineering effort: at the time of writing this paper, none of existing codebases including GhostClip and Opacus remove the forward hooks, because no analysis has established that only BiTFiT, not LoRA/Adapter/Compactor or full fine-tuning, can be activation-free. This is a simple yet effective trick to save memory in large model training. Our code will be open source to benefit the whole DP community.
>
> In summary, the algorithmic innovation that improves the naive way of combining is visualized in Figure 2: we significantly simplify the computation graph from left and upper to the DP-BiTFiT (bottom right), with complexity guarantee.
>
> 2. Why is Ci given as an input to Algorithm 1? I wonder if you mean the clipping function C.
>
> **Response** We meant Ci is initialized as 1, so that if one ignores all red lines in Algorithms 1, one can recover the non-DP SGD (which still uses Ci in line 8 because we want to write the DP/non-DP SGD in a unified way).
>
> 3. Is this any better from a clipping perspective? Do you have to materialize the per-example bias gradients in order to compute their norms? You could combine this with ghost clipping, right?
>
> **Response** No. We discussed this above Algorithm 1, in the "Novelty" paragraph that ghost clipping only works for the weight, not the bias. We have to materialize the per-sample bias gradients but this is extremely cheap. There are about 0.1% parameters in biases and materializing per-sample bias gradients only incurs $O(Bp)$ time/space complexity, in contrast to $O(BTpd)\approx O(BTp^2)$ of per-sample weight gradients. This is empirically confirmed in Figure 4, including on GPT2-large.
>
> We hope the reviewer can reconsider the score given that the complexity analysis is an important component of this work. Even though the methods are not new, a theoretical analysis could be insightful and novel, like those convergence analysis to the SGD algorithm.

---

> > ### Comment · Reviewer_tcjD · 2023-11-22
> >
> > Thank you for your comment.
> >
> > Regarding novelty: the "complexity analysis" in Appendix C seems pretty basic. (I don't think it is even strictly correct: you probably want to be using $O$ notiation since you are adding the time of different types of operations, and summation of $B$ elements can be done in fewer than $B$ addition operations.) The engineering aspect to remove the forward hooks is nice, but not really so challenging to merit publication in a top-tier conference. So while I would even say this method looks very promising and I could see it being very useful, the novelty and creativity is on the low side.
> >
> > Regarding Algorithm 1, it is confusing that $C$ is both a function and $C_i$ is the clip of a round, and $C_i$ can be input to the function. Maybe rewrite it so line 8 is different depending on whether DP is added (clips or does not clip during summation).
> >
> > I don't understand why you cannot combine with ghost clipping. The paragraph you mentioned says that they are "orthogonal", which to me sounds like they can both be used and each be beneficial. In what sense does ghost clipping "only work on the weights"? It scales the entire gradient, weights and biases alike.
> >
> > I hope you can answer these questions while the discussion period is ongoing. Because I think the method is useful (if not very original) I will provisionally raise my score to 6.

---

> ### Author Response · Authors · 2023-11-22
>
> Thank you for extending the discussion. We would like to keep the current complexity analysis without $O$ notation because 1) The complexity is computed correctly up to leading terms, e.g. if the actual complexity is 3BTp+5Bp, the leading term is cubic and leaves 3BTp. 2) The big $O$ notation is an asymptotic symbol hence not precise. In full fine-tuning, standard non-DP optimization takes 6BTpd time complexity and GhostClip takes 10BTpd, the significant slowdown is only depicted without writing both as $O(BTpd)$. 3) We agree that the **time** of B operations is not B times, due to parallelism of GPU; however, it still holds that the theoretical complexity increases by B times.
>
> We have rewritten Algorithm 1 to include your suggestion.
>
> Regarding ghost clipping, it is a technique that computes the per-sample weight gradient norm without materializing the per-sample weight gradient. It specifically leverages activation and output gradient (see Equation 3 of  https://arxiv.org/abs/2110.05679) via an algebraic trick, but the bias gradients do not need the activation (see our Equation 4; this activation-free property is key to BiTFiT). Therefore, 1) GhostClip only applies on the weights, not the bias; 2) GhostClip has time/space complexity $O(BT^2)$ making it inefficient on large $T$ tasks such as long context and high-resolution image (see Section 3.1 of https://arxiv.org/pdf/2210.00038.pdf), but per-sample bias gradients are computed with complexity independent of $T$. These two distinct differences make the computation of weight/bias gradients "orthogonal".
>
> We are happy to discuss further if needed.

---

> > ### Comment · Reviewer_tcjD · 2023-11-22
> >
> > I still think it is incorrect and misleading to count the number of "operations" without big O notation. It is not meaningful to say that "the total time complexity is $3Bp$". For example, you are considering taking a square (a multiplication) to be equivalent to summing (an addition). These operations may take different amounts of time, which is why we usually hide the constants behind the big O. Furthermore, as I tried to allude earlier, finding the sum of $n$ elements requires fewer than $n$ additions. To say "big O notation is an asymptotic symbol hence not precise" has it precisely backwards: big O conveys what is important (asymptotic behavior) while supressing what is unimportant (whether an addition or multiplication operation takes longer).
> >
> > I don't understand what you mean "the bias gradients do not need the output gradient". All of the parameters' gradients depend on the output: it's only via the output that a parameter has any influence on the loss. You even say in the paper "the output gradient is used to compute the per-sample gradient of weights and biases". What am I misunderstanding?

---

> ### Author Response · Authors · 2023-11-23
>
> We would like to clarify that it is norm to measure time complexity in float-point operation (FLOP), so that either a multiplication or an addition is one operation, despite two operations may cost different wall-clock time. Within this measure, our analysis can give precise calculation of complexity, without using big O. One particular benefit of using our notation (instead of using big O) is that we can infer the wall-clock time: e.g. standard SGD takes 6BTpd time complexity, GhostClip takes 10BTpd, standard BiTFiT takes 4BTpd, and our DP-BiTFiT takes 4BTpd+Bp. This translates to very accurate estimate of running time: GhostClip takes 1.7X more time than SGD, BiTFiT (DP and standard) takes 66% time of SGD.
>
> We can also write DP-BiTFiT complexity in different operations, e.g. (only for illustration purpose, not the actual breakdown) $1.8BTpd\*O_{multiplication}+2.2BTpd\*O_{addition}+Bp\*O_{other}$. Notice that our main conclusions will hold in either notation: for instance, 1) DP-BiTFiT is almost as fast as BiTFiT, 33% faster than full fine-tuning; 2) the overhead ($Bp\*O_{other}$) is independent of $T$, hence DP-BiTFiT is uniquely suitable for high dimension tasks.
>
> I made a typo in my previous response. Let me correct it by stating "the bias gradients do not need the activation", not the output gradient. I believe there is a misalignment of output gradient here. It is not the gradient of loss w.r.t. the last layer's output. In fact, it is the gradient of loss w.r.t. each layer's output. It would be clear to say: 1) the k-th layer's weight gradients need k-th layer's activation and k-th layer's output gradient. 2) the k-th layer's weight gradients need only k-th layer's output gradient. Therefore, GhostClip does not apply to weight gradients because it explicitly works with the outer product of activation.

---

> > ### Comment · Reviewer_tcjD · 2023-11-23
> >
> > Thank you for your response.

---

> > > ### Author Response · Authors · 2023-11-23
> > >
> > > No problem. Glad to have this discussion.

---

### Official Review · Reviewer_BrY1 · 2023-10-29

**Soundness:** 3 good
**Presentation:** 1 poor
**Contribution:** 3 good
**Rating:** 6
**Confidence:** 3

**Summary:**

This paper studies the differentially private fine-tuning of large pre-trained models. The key novelty of the paper is that the authors show that fine-tuning only the bias term can match or even outperform the SOTA DP fine-tuning algorithm in different tasks. By such bias-term fine-tuning idea, the number of trainable parameters can be even smaller than previous parameter-efficient fine-tuning methods (e.g., LORA), and faster and more memory-efficient than the full-parameter DP fine-tuning.

**Strengths:**

1. The paper presents the key idea with detailed and persuasive motivation.
2. The idea proposed in this paper can bring significant advantages in reducing computation costs while improving model utility.
3. The authors demonstrate relatively comprehensive experiments to support the claim advantages, including text classification, natural language generation, and image classification.

**Weaknesses:**

The paper is pretty much self-contained, there may be only some concerns on how general this method can be.
1. The paper can provide more intuitions about why fine-tuning the bias term can be effective enough with pre-trained models. As a comparison, LoRA is supported by a strong intuition that fine-tuning updates can be considered low-ranks.
2. The limitation of the proposed algorithm may deserve some more discussion. While remark 4.2 mentioned that the method may be less effective for models with convolutional layers without bias terms, it is not clear whether all modules' bias terms (attention, fully connected, convolutional, etc.) have the same effect or some of them are more important than others. Besides, as some recent LLM architecture may not include biases in some layers (PaLM [0]) to increase training stability, it is unclear how general the proposed method can be in practice.


[0]Chowdhery, Aakanksha, et al. "Palm: Scaling language modeling with pathways." arXiv preprint arXiv:2204.02311 (2022).

**Questions:**

1. Is there any potential limitations of the proposed algorithm in terms of the fine-tuning tasks? Can we expect similar improvement if we fine-tune for more complicated tasks, for example fine-tune LLaMA for GSM8k or for MMLU tasks?
2. Is there any requirement or conclusion about how well-pre-trained a model should be or how complex a model should be so that the proposed method can be effective? An inappropriate extreme example might be that one should not expect tuning the bias in a linear model to fit arbitrarily shifted distribution.
3. Maybe related to the above question, is there intuition to support the proposed method?
4. Are the bias terms in different modules (attention, fully connected, convolutional, etc.) have the same effects?

---

> ### Author Response · Authors · 2023-11-21
>
> We thank the reviewer for the comment and the positive feedback. Here is a point-to-point response.
>
> 1. The paper can provide more intuitions about why fine-tuning the bias term can be effective enough with pre-trained models. As a comparison, LoRA is supported by a strong intuition that fine-tuning updates can be considered low-ranks.......is there intuition to support the proposed method?
>
> **Response** We agree more understanding of why BiTFiT works is desirable. This has been difficult even for the original paper in the non-DP regime. Looking backwards from the effectiveness of BiTFiT, we may have a intuition that the fine-tuning updates are generally in low-dimension, as is also the case for LoRA (despite the rank, the number of trainable parameters is small). We may test this hypothesis by randomly selecting a small number of parameters to update at each layer. However, even if any small number of parameters would work, BiTFiT has its unique advantage of being activation-free.
>
> 2. While remark 4.2 mentioned that the method may be less effective for models with convolutional layers without bias terms, it is not clear whether all modules' bias terms (attention, fully connected, convolutional, etc.) have the same effect or some of them are more important than others. Besides, as some recent LLM architecture may not include biases in some layers (PaLM [0]) to increase training stability, it is unclear how general the proposed method can be in practice.......Are the bias terms in different modules (attention, fully connected, convolutional, etc.) have the same effects?
>
> **Response** We haven't looked into a finer-grained analysis of bias terms. This could be an interesting future work had more people become interested in our method. We agree that not all models are suitable for DP-BiTFiT, but a long list of models in our Table 1 should be. Additionally, we study DP-BiTFiT as a standalone method for fair comparison to other standalone methods. In practice, one can always combine two parameter-efficient methods, such as DP-BiTFiT and last-layer training in Section 4.3. Specifically, turning on DP-BiTFiT only needs one line of code: [param.requires_grad_(False) for name, param in model.named_parameters() if 'bias' in name].
>
> 3. Is there any potential limitations of the proposed algorithm in terms of the fine-tuning tasks? Can we expect similar improvement if we fine-tune for more complicated tasks, for example fine-tune LLaMA for GSM8k or for MMLU tasks?
>
> **Response** I am experimenting with LLaMA and do observe similar behavior.
>
> 4. Is there any requirement or conclusion about how well-pre-trained a model should be or how complex a model should be so that the proposed method can be effective? An inappropriate extreme example might be that one should not expect tuning the bias in a linear model to fit arbitrarily shifted distribution.
>
> **Response** Our experiments in Table 3,4,5 show that DP-BiTFiT tends to be more comparable to full fine-tuning when model sizes are larger. This also implies that a better pre-trained model can benefit from DP-BiTFiT more: tuning only bias terms from scratch is not going to work.

---

> > ### Comment · Reviewer_BrY1 · 2023-11-22
> >
> > Thanks for the response from the authors. I leave my score as it is now, as the authors partially agree that my concerns may need more effort to be resolved in the future.

---

### Official Review · Reviewer_3Kry · 2023-10-29

**Soundness:** 3 good
**Presentation:** 3 good
**Contribution:** 2 fair
**Rating:** 5
**Confidence:** 4

**Summary:**

This paper studies private fine-tuning of large models using a parameter-efficient method. This work proposes DP-BitFit, differentially private bias-term fine-tuning, which updates bias terms during training under the DP-SGD framework.

While updating full parameters in DP-SGD would increase memory consumption and slow down the training process, DP-BitFit only needs to update 0.1% of the parameters, making it much faster than updating all parameters. This work supports this claim by providing both the time and space complexity and experimental comparisons. Another advantage of DP-BitFit is that it does not require a forward hook.

This work conducts experiments on both NLP tasks and computer vision (CV) tasks, considering privacy constraints within the ranges of [3, 8] for NLP and [1, 8] for CV.

**Strengths:**

1. The paper is well-motivated and easy to follow. Specifically, the theoretical analysis of space and time complexity of different methods is comprehensive.

2. The experiments include both cv and nlp tasks, that are two of the main applications of foundation model use. The evaluation is comprehensive and ablates the model size, mode architectures, privacy levels.

**Weaknesses:**

1. My main concern with this paper is the novelty of this method. It appears that this method directly adapts the existing method BitFit to the DP-SGD.

2. The performance of DP-BitFit is limited in some scenarios and requires the additional design for two phases, which makes the results of DP-BitFit less significant.

Minors:
1. Presentation issues: The table number and title should appear before the table (Table 7- 16). It would be better to be more careful with the use \citep and \citet.

**Questions:**

1. In Table 4, it is somewhat weird to me that the perplexity of non-private results is worse than private results in several cases. Also, Table 4 and Table 13 show that DP-BitFiT on GPT2-large is better than DP-BiTFiT is better than DP-SGD(full), I wonder if this is due to the dimensionality issue in DP-SGD, or sub-optimal hyperparameters for DP-SGD (full), or if it is because the comparison is not averaged across several runs for statistical significance.

2. Comparison with DP parameter efficient methods for efficiency. The authors provide the theoretical comparison of DP-BitFit with other DP parameter-efficient methods. Figure 3 and Figure 4 compared the memory, speed, maximum throughput and batch size for DP-BitFit and full parameter updating methods. It would be better to also include the comparison to other DP parameter efficient methods such as DP-LoRA.

---

> ### Author Response · Authors · 2023-11-21
>
> We thank the reviewer for the comment. We would like to clarify a misunderstanding in the "Weakness". **We do not require "the additional design for two phases."** This was an alternative method presented in older version of this paper, hence it should not be treated as a main part of this work on which the review is based.
>
> 1. Regarding the novelty, we claim at least 2 types of new contributions: 1) Complexity analysis of DP parameter-efficient fine-tuning (PEFT): DP-BiTFiT in main text, DP-LoRA and so on in appendix C. This is a missing piece in previous DP PEFT literature and significantly helpful in determining the benefit of applying different PEFT methods. E.g. we rigorously show the complexity saving is 50%, and the benefit of BiTFiT on long-context language tasks or high-resolution image tasks. This analysis is also missing in the non-DP BiTFiT. 2) Engineering effort: at the time of writing this paper, none of existing codebases including GhostClip and Opacus remove the forward hooks, because no analysis has encouraged to do so. Note our code will be open source to benefit the whole DP community.
>
> 2. Regarding Table 4, we believe the situation is natural: the perplexity is only one of many performance measures other than BLEU. In fact, it is not unusual sometimes the cross-entropy loss (i.e. perplexity) is at odds with the performance measure, e.g. in \url{https://arxiv.org/pdf/2106.07830.pdf} Figure 8. We are not sure why DP-BiTFiT seems to outperform DP-full on GPT2-large, though we do consistently observe that the gap is smaller as we scale up the model sizes. We have tuned the hyperparameters of DP-full carefully so our hypothesis is the dimensional issue.
>
> 3. We haven't compared to DP-LoRA in all experiments. We are happy to do so in the camera-ready revision if the reviewer can consider raising the score.

---

> ### Author Response · Authors · 2023-11-22
>
> Dear reviewer,
>
> We have addressed your comments point-by-point, trying our best to clarify any confusion. We have changed all \cite to \citep in this revision and corrected the table captions.
>
> Specifically, **we do not need two-phase training in this paper** (it's merely an alternative in the appendix, hence should not be counted towards *Weakness*). We sincerely hope you would reconsider your score if the score is based on the mis-understanding.
>
> We would love to elaborate the significance of our complexity analysis and in the engineering efforts: removing the forward hooks is **never implemented in previous codebases** because only DP-BiTFiT is activation-free, not DP-LoRA or any other methods. Being activation-free is an important characteristics that make DP-BiTFiT uniquely suitable for high dimension (e.g. long context or high resolution image) tasks, which was not discovered in existing DP parameter-efficient fine-tuning literature which lacks the analysis on the computation graph and complexity.
>
> Your evaluation is important to us and we look forward to further discussion if needed.

---

> > ### Comment · Reviewer_3Kry · 2023-12-04
> > **Thank you for your response**
> >
> > Thank you for your response. I have carefully reviewed the authors' rebuttal and the feedback from other reviewers.
> >
> > 1. For the W2 "two-phase training", I am more curious about when DP-BitFit would be preferred in terms of utility. The design of two-phase training is motivated as DP-BitFit does not work well in several scenarios. Therefore, my concern is more about the utility and practical use of DP-BiTFit. For example, if someone is able to train the model in the full parameter set-up, maybe it is more straightforward to train the model with full parameter instead of the two-phase training to use DP-BitFit. To summarize, for W2, it is not clear to me the effectiveness of DP-BitFit in terms of utility. I find that reviewer oSFo and reviewr regQ also share this question.
> >
> > 2. W1. I still share the concern of novelty of this work with other reviewers.
> >
> > 3. Q1. Thanks for your explanation of the evaluation metric. Thank you for reporting several metric for the experiments. It is unclear to me that this is because a single metric may not be fully reliable for evaluation even in the non-private set-up or it is because the noise added in training for DP may serve as a regularization term in the optimization process. For the results on GPT2-large, it would be much appreciated if the authors could provide more insights on the results as suggested by reviewer oSFo.
> >
> > 4. Q2. Maybe my Q2 is more out of curiosity for the significance of the efficiency analysis in practice. Though the theoretical analysis shows that DP-BitFit enjoys the benefits in terms of efficiency, in practice, it is not clear to me such advantage when compared to other parameter efficient training method is still significant . Therefore I cannot promise I would increase the score for the experiments before the results. After reading other reviewer's comment, I think conducting more analysis in terms of the utility is more desirable.
> >
> > Taking all this into consideration, I decide to keep my score .

---

### Official Review · Reviewer_regQ · 2023-11-06

**Soundness:** 2 fair
**Presentation:** 3 good
**Contribution:** 2 fair
**Rating:** 3
**Confidence:** 4

**Summary:**

The paper addresses the problem of differentially private (DP) fine-tuning of large pre-trained models. Existing research has shown that high accuracy can be achieved under strong privacy constraints, but it often comes at the cost of significant computational overhead or modifications to the network architecture.

The authors propose  "differentially private bias-term fine-tuning" (DP-BiTFiT)  to strike a balance between accuracy and efficiency. DP-BiTFiT achieves state-of-the-art accuracy levels for DP algorithms while maintaining the efficiency of standard BiTFiT (fine-tuning without privacy constraints). The efficiency enables the application of DP fine-tuning to language and vision tasks involving long-sequence texts and high-resolution images, which were previously computationally challenging using existing methods.

**Strengths:**

Efficiency and Scalability: The paper presents an efficient and scalable approach to differentially private fine-tuning, which is crucial for handling large models and complex tasks. Compared with existing parameter-efficient DP fine-tuning, DP-BiTFiT is model-agnostic, i.e., it does not require modifications to the network architecture. The paper demonstrates that DP-BiTFiT outperforms DP full fine-tuning in terms of speed and memory usage, even surpassing the efficiency of standard full fine-tuning.

Practical Applicability: The ability to conduct DP fine-tuning on language and vision tasks with long-sequence texts and high-resolution images expands the practical applications of privacy-preserving machine learning.

**Weaknesses:**

**Limited contribution** Although the paper emphasizes that DP-BiTFiT is not merely adding differential privacy to an existing method (BiTFiT), it is hard to find much evidence for this point. Removing the forward hook is quite natural as one does not have to compute the gradient on weights. This is a natural choice when combining BiTFiT and differential privacy. Given existing results of DP full fine-tuning and parameter-efficient fine-tuning, the contribution of this paper is rather incremental.

Moreover, the empirical evaluation shows performance drop of DP-BiTFiT compared with other parameter-efficient fine-tuning techniques.
The paper mentions that DP-BiTFiT is efficient for a wide range of tasks, but it would be helpful to provide specific examples and use cases to illustrate its versatility.

**Questions:**

1. In the Contribution 4, "DP-BiTFiT is a unique algorithm in that the computation overhead is independent of the feature dimension T" where T is the sequence length. The author should be more specific for this claim.

**Details Of Ethics Concerns:**

The paper provides new algorithms for private machine learning with differential privacy guarantee, which could be used to protect privacy in machine learning pipeline.

---

> ### Author Response · Authors · 2023-11-21
>
> We thank the reviewer for the comment. Here is a point-to-point response.
>
> 1. Removing the forward hook is quite natural as one does not have to compute the gradient on weights. This is a natural choice when combining BiTFiT and differential privacy.
>
> **Response:** We agree it is natural to remove the forward hooks *with the benefit of hindsight*. But being natural does not conflict the significance of our choice. At the time of writing this paper, none of existing codebases including GhostClip and Opacus remove the forward hooks, because no analysis has encouraged to do so. We are the first to point this rigorously with complexity analysis and computation graph in Figure 2. We also point out that even the original non-DP BiTFiT paper does not investigate these aspects, e.g. the complexity saving is 50%, and the benefit of BiTFiT on long-context language tasks or high-resolution image tasks. Besides, we complement the missing piece of complexity analysis in the existing literature of DP parameter-efficient fine-tuning (PEFT), e.g. DP-LoRA, making the comparison among methods more solid in theory. We believe this is good practice that can benefit the community and provide this in Appendix C.
>
> 2. Moreover, the empirical evaluation shows performance drop of DP-BiTFiT compared with other parameter-efficient fine-tuning techniques.
>
> **Response:** We agree that there is sometimes a minor drop of DP-BiTFiT, but overall DP-BiTFiT is comparable when applied alone. For fair comparison, we study DP-BiTFiT as a standalone method to compare with other PEFT. A promising application is actually to apply DP-BiTFiT together with LoRA or last-layer tuning, as we have shown in Section 4.3. Also, a small performance drop is acceptable in the cases when we emphasize the memory efficiency (since DP-BiTFiT is activation-free) and implementability (only need one line of code $$[param.requires\_grad\_(False) for name, param in model.named\_parameters() if 'bias' in name]}$$).
>
> 3. In the Contribution 4, "DP-BiTFiT is a unique algorithm in that the computation overhead is independent of the feature dimension T" where T is the sequence length. The author should be more specific for this claim.
>
> **Response:** We have explained the computation overhead of DP-BiTFiT in the last column of Table 2, where both the time and the space complexity is O(Bp), independent of T. This is also empirically verified in Section 3.3: we quote "DP-BiTFiT is amazingly scalable since its *computational overhead* is negligible and independent of T (though the *total complexity*, mainly due to forward and output gradient, is still linear in T)."
>
> We hope the reviewer can raise the score if satisfied.

---

> > ### Author Response · Authors · 2023-11-22
> >
> > Dear reviewer,
> >
> > We have addressed your comments point-by-point, trying our best to clarify any confusion. We hope you would agree that **removing the forward hook is natural, but being natural does not mean it is insignificant or not novel**, because the trick is effective and supported by complexity theory and experiments, but never adopted before. Similar to Charles Steinmetz's anecdote, it is not about how easy to make the change, it's about knowing where to make the change.
> >
> > Your review is important to us. We sincerely hope you would reconsider the score.

---

### Official Review · Reviewer_oSFo · 2023-11-06

**Soundness:** 3 good
**Presentation:** 3 good
**Contribution:** 4 excellent
**Rating:** 8
**Confidence:** 3

**Summary:**

This work studies a differentially private version of bitfit, termed DP-Bitfit. DP-bitfit, like bitfit, works by tuning only the bias terms in the model. However, the key innovation here is recognizing that bitfit is highly parameter efficient which is crucial for DP learning. Second, is recognizing that the gradients are much computationally cheaper to calculate due to it being activation-free. Overall, this work shows impressive empirical results evaluated on both image and text models.

**Strengths:**

Overall, this work shows to be a strong submission. This work includes comprehensive comparison between different existing methods (e.g., ghost clipping) and shows impressive benefits in both memory, throughput, and final model utility. It is perhaps unsurprising that this performs so well given the performance of non-DP bitfit, however, the core benefit of this work is recognizing its potential for the DP setting.

The organization of the work is clear, and the work includes several key figures and diagrams that help the reader follow the work. For example, Figure 1 clearly shows the empirical benefits of the approach and Section 3/ Figure 2 the  asymptotic benefits.

Though the novelty is lower because this is essentially applying DP-SGD to the existing "bitfit" algorithm, this work include smany empirical results evaluated across different model families (e.g., ViT models,resnets, and roberta models) on both text and image classification as well as text generation.

**Weaknesses:**

This works lacks a clear empirical exploration of the difference between the activation and the model memory. Right now, it is clear that the algorithm uses much less memory and compute. However, it is unclear how the total memory in figure 1 is split between storing the model, materializing activations, etc.

Figure 4 is extremely confusing. How can DP-BITFIT be both on the x-axis and in the legend? What does maximum through of algorithm mean?

**Questions:**

In table 5, how were the results of Bu et al. obtained? I could not find any 98.9% performance (Cifar-10 result) in their work.

---

> ### Author Response · Authors · 2023-11-21
>
> We thank the reviewer for the comment and appreciating our work. Although in Figure 1 we didn't precisely split the memory between storing the model, materializing activations, etc., this information can be inferred (approximately in the middle sub-plot): given that DP compactor/LoRA/Adapter have similar (though larger) number of trainable parameters to DP-BiTFiT, we claim most of the memory difference is indeed in the activations.
>
> We would like to clarify Figure 4. The x-label and legend are both correct. The DP-BiTFiT is serving as a reference line diagonally. Any method such as "non-DP full" can be comparable to DP-BiTFiT by looking at the slope (less steeper means less scalable). Each dot represent the actual throughput or memory (from left to right is large to small models). The throughput means number of examples processed per second by maximizing the batch size, as a measure of training speed. We will clarify it in the camera-ready revision.
>
> We are grateful that the reviewer spot this typo in Table 5: the vit-large result is Ours, obtained through full fine-tuning. The horizontal line should be one row higher.

---

> > ### Comment · Reviewer_oSFo · 2023-12-04
> > **Remain mostly positive.**
> >
> > I have read the rebuttal and remain mostly positive above this work.
> >
> > First, let me thank the authors for the clarifications, these make sense now.
> >
> > I have also read the other reviews and their rebuttals. I agree with the other reviewers about the lack of novelty but still believe that the empirical results and learning remain quite interesting and deserve merit. Even if not algorithmically novel, this work does provide an interesting new direction for DP-learning in parameter/memory constrained settings.
> >
> > However, upon reconsideration, I do believe that this work could be improved more before meeting the bar of a top-tier conference. In particular, if the main novelty/contribution of this work is the direction and empirical findings, this work could do a better exploration there. The most important improvement here is, in my opinion, answering the question: "when should a practitioner use DP-BitFit instead of existing methods" or analogously, "when is DP-BitFit comparable/better". In terms of compute, there seems to be a clearer answer, but in terms of utility, this story remains, or is even made more confusing after this work. It is unclear under what settings of the problem parameters (model size, task difficulty, number of example, etc.) dp-bitfit will outperform, perform on par, or perform worse than existing methods. And so, at best this work adds yet a new method for dp fine-tuning that is not well understood. This could be improved through both additional experiments exploring how the problem parameters impact the tuning method and through more discussion and interpretations of the existing results.
> >
> > Overall, I will leave my score the same, but, not advocate strongly for the acceptance.

---

### Meta-Review · Area_Chair_MR9K · 2023-12-06

**Metareview:**

This paper studies the question of differentially private (DP) finetuning of large models. The approach is to adapt the bias-term fine-tuning (BiTFiT) algorithm (Zuken et al., 2020) which freezes most of the model and, for the non-frozen part, trains only the bias term. This paper adapts the algorithm by applying the DP-SGD algorithm (i.e. clip & add noise) of (Abadi et al., 2016). The authors then observe that backpropagation for DP-BiTFiT does not require the activation tensors from the forward propagation because only the bias terms are trained. By removing these activation tensor caching, the authors show significant improvements in terms of running time and memory. Meanwhile, the accuracy remains roughly similar (within 1% most of the time) of SOTA DP finetuning results.

## Strengths

- It is nice to know that BiTFiT works so well with DP, and this might have practical applications.

## Weaknesses

- In hindsight, the finding is not very surprising given that (non-private) BiTFiT itself is very parameter-efficient and it is well known that parameter-efficient training usually works well with DP.

- The way DP is incorporated into BiTFiT is a trivial application of DP-SGD.

- The memory & running time improvement in this paper seems to be more of an engineering trick rather than a significant advancement in understanding of private ML. (And the trick seems quite trivial too, although maybe this is partially by benefit of hindsight.)

**Justification For Why Not Higher Score:**

As stated above, the main novelty is more of an engineering trick rather than a significant insight into private ML. Although some papers with similar "tricks" have appeared in top conferences before (the [Subramani et al., NeurIPS 2021] paper comes to mind), those typically apply to a large class of algorithms and offers more significant improvements. Meanwhile, the trick in this paper only applies for only a single method of finetuning (BiTFiT). Overall, I think it would be more appropriate to publish this at a more specialized venue (on practical DP ML) rather than at ICLR.

**Justification For Why Not Lower Score:**

N/A

---

### Decision · Program_Chairs · 2024-01-16

Reject